# A multimodal iPSC platform for cystic fibrosis drug testing

Andrew Berical[1,2], Rhianna E. Lee [3,4,10], Junjie Lu[5,10], Mary Lou Beermann[1], Jake A. Le Suer[1], Aditya Mithal [1], Dylan Thomas[1], Nicole Ranallo[1], Megan Peasley[5], Alex Stuffer[5], Katherine Bukis[5], Rebecca Seymour[5], Jan Harrington[5], Kevin Coote[5], Hillary Valley[5], Killian Hurley [6,7], Paul McNally [8,9], Gustavo Mostoslavsky [1], John Mahoney [5], Scott H. Randell [3,4] & Finn J. Hawkins [1,2✉]

Cystic fibrosis is a monogenic lung disease caused by dysfunction of the cystic fibrosis transmembrane conductance regulator anion channel, resulting in significant morbidity and mortality. The progress in elucidating the role of CFTR using established animal and cell-based models led to the recent discovery of effective modulators for most individuals with CF. However, a subset of individuals with CF do not respond to these modulators and there is an urgent need to develop novel therapeutic strategies. In this study, we generate a panel of airway epithelial cells using induced pluripotent stem cells from individuals with common or rare CFTR variants representative of three distinct classes of CFTR dysfunction. To measure CFTR function we adapt two established in vitro assays for use in induced pluripotent stem cell-derived airway cells. In both a 3-D spheroid assay using forskolin-induced swelling as well as planar cultures composed of polarized mucociliary airway epithelial cells, we detect genotype-specific differences in CFTR baseline function and response to CFTR modulators. These results demonstrate the potential of the human induced pluripotent stem cell platform as a research tool to study CF and in particular accelerate therapeutic development for CF caused by rare variants.

[1] Center for Regenerative Medicine of Boston University and Boston Medical Center, Boston, MA 02118, USA. [2] The Pulmonary Center and Department of Medicine, Boston University and Boston Medical Center, Boston, MA 02118, USA. [3] Marsico Lung Institute and Cystic Fibrosis Research Center, Department of Cell Biology and Physiology, University of North Carolina at Chapel Hill, Chapel Hill, NC 27599, USA. [4] Department of Cell Biology and Physiology, University of North Carolina at Chapel Hill, Chapel Hill, NC 27599, USA. [5] Cystic Fibrosis Foundation, Lexington, MA 02421, USA. [6] Department of Medicine, Royal College of Surgeons in Ireland, Education and Research Centre, Beaumont Hospital, Dublin, Ireland. [7] Tissue Engineering Research Group, Royal College of Surgeons in Ireland, Dublin, Ireland. [8] RCSI University of Medicine and Health Sciences, Dublin, Ireland. [9] Children's Health Ireland, Dublin, Ireland. [10]These authors contributed equally: Rhianna E. Lee, Junjie Lu. ✉email: hawk@bu.edu

Cystic fibrosis (CF) is caused by variants of the cystic fibrosis transmembrane conductance regulator (CFTR) gene and leads to multi-organ disease particularly affecting the respiratory system. The CFTR gene encodes an anion channel involved in the regulation of proper airway surface liquid hydration, viscosity, and pH[1]. CFTR variants lead to abnormally viscous secretions in the airways which cause sino-pulmonary infection, bronchiectasis, and a shortened life span[2]. More than 2000 variants in the CFTR gene have been identified to date, with several hundred known to cause clinical disease. Variants are classified according to the molecular defects they cause, including variation in protein quantity (classes 1, 5), trafficking (class 2), channel opening (class 3), and ion conductance (class 4)[3]. Due to the range and combinations of CFTR variants, targeted pharmacotherapy approaches are complex. Depending on their CFTR variant, ~90% of individuals with CF should benefit from a newly developed class of medications termed CFTR modulators which improve protein trafficking, function, and patient-centered clinical outcomes[4–9]. However, CFTR modulators are not curative, expensive, required in combination therapy, and theoretically are required life-long. Furthermore, individuals with CF due to class 1 variants lack targeted therapies and require urgent and ambitious approaches to ameliorate their disease.

Preclinical in vitro models were critical to the discovery and approval of CFTR modulators and will almost certainly play a central role in advancing therapeutic options for CF further. Many cellular models of CF exist, including heterologous cell lines, primary human rectal organoids, human nasal epithelial cells, and human bronchial epithelial cells (HBECs), each with its own advantages and disadvantages that have been thoroughly reviewed elsewhere[10]. On one end of the spectrum, heterologous cell lines transduced with CFTR (e.g., Fischer Rat Thyroid cells) have enabled high-throughput screening approaches that led to the identification of CFTR modulators but bear little resemblance to the human tissues affected by CF[11]. On the other end of the spectrum are HBECs: the current gold-standard cell-based platform for the preclinical assessment of CFTR-mediated ion transport. Obtained from either bronchoscopic biopsy or explanted lungs, HBECs from individuals with CF differentiate in air-liquid interface (ALI) culture into a pseudostratified airway epithelium with morphologic, molecular and physiologic similarities to the in vivo human airway[12,13]. Electrophysiologic studies (e.g., Ussing chamber) of CFTR-dependent ion flux in CF HBEC ALI cultures are sensitive and predictive of in vivo response to modulators[10]. The efficacy of a candidate drug is typically validated in HBECs prior to advancing to clinical trials. From this in vitro pipeline, there are now several FDA-approved CFTR modulators. VX-770, a potentiator that increases the probability of channel opening was the first to be approved for the class 3 gating variant, Gly551Asp[8]. Clinical treatment with VX-770 leads to a 10% $FEV_1$ increase, decreased exacerbations, among other patient-centered outcomes. The CFTR corrector molecules VX-809 and VX-661, each used in combination with VX-770 were approved for patients harboring two copies of Phe508del, the most common CF-causing variant, though the clinical efficacy has been modest (2.6–4% $FEV_1$ increase)[5,9]. A second-generation corrector (VX-445) was recently approved for both homozygous and heterozygous Phe508del individuals and shows promising clinical improvement in combination with VX-661 and VX-770 in a large randomized clinical trial (14% $FEV_1$ increase)[7]. In addition, data from preclinical platforms recently formed the basis to expand the approval of CFTR modulators to patients with rarer CFTR variants not represented in clinical trials[14,15].

While the clinical response to CFTR modulators is encouraging, currently approved drugs do not afford a benefit to ~10% of all individuals with CF due to underlying CFTR variants. Most notably, patients with class 1 nonsense variants (e.g., G542X, W1282X) have severe CF without targeted therapeutic options[16,17]. Compared to the class 2 (Phe508del) and 3 (Gly551Asp) variants above, class 1 variants are fundamentally different; premature termination codons (PTCs) within mRNA transcripts cause mRNA degradation by nonsense-mediated decay (NMD) or cause premature translation termination, leading to absent or truncated non-functional protein. A successful therapeutic strategy for these individuals will likely involve complex pharmacotherapy, gene-editing, gene-delivery, or cell-based approaches. From a pharmacotherapy approach, combinatorial regimens using read-through molecules (e.g., G418), inhibitors of nonsense-mediated decay (e.g., SMG1i), and novel modulators are being developed[18,19]. The development of these ambitious approaches would likely be accelerated by unrestricted access to patient-specific airway cells, however, there is a major bottleneck in obtaining HBECs from individuals with rare class 1 CFTR variants and electrophysiologic measures of CFTR-dependent current are relatively low-throughput. As a result, other in vitro platforms and CFTR functional assays are being developed to overcome these hurdles. For example, rectal organoids are quite readily obtained via a minimally invasive biopsy from individuals with CF and can generate a near limitless supply of patient-specific cells[10,20,21]. After the rectal biopsy, CFTR-expressing rectal organoids are cultured and CFTR function is measured using the forskolin-induced swelling (FIS) assay. Activation of the CFTR channel is initiated by adding forskolin (± CFTR modulator) which leads to a CFTR-dependent increase in organoid size within hours. This platform detected CFTR modulator effects in cells from individuals with different CFTR variants and correlated with in vivo drug effects[20–22].

Human-induced pluripotent stem cells (iPSCs) have several properties that make them promising candidates for advancing CF therapeutics. iPSCs contain the complete and unique genetic code of the donor, including specific CFTR sequence. They are routinely generated by the "reprogramming" of patient-derived somatic cells (e.g., dermal fibroblasts, PBMCs) through the overexpression of OCT3/4, SOX2, KLF4, and C-MYC[23,24]. By recapitulating key embryological events, iPSCs can be differentiated into alveolar and airway epithelial cells[25–36]. We previously applied these protocols for proof-of-concept CF disease modeling; for example, we corrected the CFTR sequence in Phe508del iPSCs and measured CFTR-dependent airway as well as intestinal epithelial spheroid swelling in FIS assays[33,37]. We recently published the successful derivation of airway basal cells, the major stem cell of the airway epithelium, from iPSCs. These iPSC-derived airway basal cells ("iBCs") self-renew in vitro and under similar ALI culture conditions used for HBECs, differentiate into a mucociliary epithelium that is morphologically and transcriptionally similar to primary HBEC ALI cultures[29]. iBC-derived ALI cultures from non-CF individuals expressed CFTR and in Ussing chambers exhibited CFTR-mediated transepithelial ion flow at a magnitude expected of non-CF HBECs.

In this study, we hypothesize that iPSC-derived CFTR-expressing airway cells from individuals with class 1–3 CFTR variants can be used to measure baseline CFTR function as well as rescue in response to CFTR modulator treatment. We adapted two established CFTR functional assays: (1) FIS of 3-D airway epithelial cell spheroids and (2) electrophysiologic measurement of CFTR-dependent current across mucociliary ALI cultures, both of which detect responses to clinically approved CFTR modulators in a genotype-dependent manner. Furthermore, combinatorial treatment of iPSC-derived airway cells from individuals homozygous for class 1 variants (W1282X or G542X) led to a significant increase in CFTR function. This iPSC-based platform can overcome a major limitation in the development of

therapeutic strategies for CF, particularly for individuals with rare variants by providing a near limitless supply of patient-specific cells with the ability to quantify CFTR function.

## Results

**CF iPSCs efficiently differentiate into airway epithelial spheroids.** In order to develop a standardized approach for the study of multiple classes of *CFTR* variants, we first sought to establish a bank of iPSCs from individuals with CF. We assembled a panel of nine iPSC lines from individuals with class 1–3 variants in the *CFTR* gene (Fig. 1A). Reprogramming of iPSCs from somatic cells was performed as previously described (Supplementary Fig. 1)[23,24,38–40]. We preferentially focused on identifying individuals with homozygous variants to exclude the potential confounding effect of compound heterozygous variants from different classes that might obscure the interpretation of CFTR modulator responses. For class 2 variants, a total of three iPSC lines were studied. Two lines were homozygous for the Phe508del variant (c.1521_1523delCTT, p.Phe508del), one of which was newly reprogrammed (iPSC clone: "CFTR4-2", hereafter Phe508del #1), and the other was previously described (iPSC clone: "RC2 204", Phe508del #2). A third iPSC line was derived from an individual with compound heterozygous variants Phe508del/Ile507del (p.Ile507del, c.1519_1521delATC), (iPSC clone: "C17", hereafter Phe508del #3) (see "Methods" for cell line details)[39,42]. For class 1, two iPSC lines were previously generated which were homozygous for a nonsense variant in exon 23 (c.3846G>A, p.Trp1282X), hereafter referred to as W1282X #1–2. We also gene-edited both alleles of the "C17" line to correct the Phe508del/Ile507del mutation and insert exon 12 variant (c.1624G>T, p.Gly542X); hereafter G542X. For class 3 variants, we identified two of the few documented individuals homozygous for the Gly551Asp variant (c.1652G>A, p.Gly551Asp). We generated an iPSC line from one, hereafter G551D #1. The second (G551D #3) was provided by the laboratory of Dr. Christine Bear at the University of Toronto/Hospital for Sick Children. Similar to the above, we generated a third class 3 iPSC line through correction of the "C17" cell line and insertion of the c.1652G>A mutation into both CFTR alleles (see Methods) (hereafter G551D #2). We also selected three established non-CF PSC lines as controls ("RUES2"; "BU1"; "BU3"), hereafter non-CF #1–3, respectively (https://www.bu.edu/dbin/stemcells)[41]. One non-CF ("BU3," non-CF #3) and one CF line ("C17," Phe508del #3) had previously undergone a gene-editing strategy to insert an NKX2-1:GFP fluorescent reporter, to enable tracking and purification of lung progenitors[28,42]. In addition, non-CF #3 contained a TP63:TdTomato reporter, while G542X and G551D #2 contained a TP63:mCherry reporter, which were used to develop the methodologies to derive airway basal cells (see "Methods")[29]. The gene-editing of the "C17" iPSC line allowed for the study of class 1, 2, and 3 mutations in the same genetic background. For each PSC line, standard quality control included G-band karyotyping and assessment of pluripotency markers (Supplementary Fig. 1); all clones had a normal karyotype.

CF and non-CF PSCs were differentiated into 3-D airway epithelial spheroids as previously described (Fig. 1B)[28,33]. The airway epithelial cell-directed differentiation protocol consists of four stages that recapitulate major developmental milestones; (stage 1, day 0–3) definitive endoderm induction, (stage 2, day 3–6) anterior foregut patterning, (stage 3, day 6–15) lung specification, and (stage 4, day 15–30) airway patterning[33]. In an effort to achieve consistent differentiation in all 12 PSC lines a number of quality control metrics were used. For all lines, we tested the efficiency of definitive endoderm induction in stage 1 by measuring the percentage of cells co-expressing the surface markers C-KIT and CXCR4 by flow cytometry[28,43]. All lines generated definitive endoderm with high efficiency (90.8–97.6% C-KIT+/CXCR4+) (Supplementary Fig. 2). To assess the efficiency of lung specification after stage 3 of the directed differentiation, we quantified the percentage of cells expressing NKX2-1 on day 14–16 by flow cytometry and similar to previous reports, observed variable NKX2-1 expression (11.7% ± 0.9%, mean ± SEM; range 2.6–29%) (Fig. 1C and Supplementary Fig. 2). To overcome this variable efficiency, we used a cell surface marker sort strategy (CD47hi/CD26neg or CPM) to enrich for NKX2-1+ cells (Fig. 1B, C)[28,44]. Sorting CD47hi/CD26neg cells significantly enriched the differentiation for NKX2-1+ cells in all lines (83.7 ± 2.2%, mean ± SEM) (Fig. 1C). In the case of the NKX2-1:GFP reporter lines (non-CF #3, Phe508del #3, G551D #2, G542X), NKX2-1GFP+ sorting was employed in place of the CD47hi/CD26neg strategy (Supplementary Fig. 2). In stage 4 of the differentiation protocol, NKX2-1+ lung progenitors were plated in 3-D culture in a media composed of FGF2, FGF10, dexamethasone, cyclic AMP, 3-isobutyl-1-methylxanthine, and Y-27632, hereafter "FGF2 + 10 + DCI + Y" (Fig. 1B). We previously reported that in the absence of exogenous activators of WNT signaling, NKX2-1+ lung progenitors formed epithelial spheroids composed of cells which express proximal airway markers including *SOX2* and *TP63* or *SCGB3A2*[33,45]. These spheroids were previously characterized extensively and determined to be composed of airway basal-like and immature secretory-like cells[45]. Expression of canonical airway epithelial cell markers in CF iPSC-derived spheroids in 3-D culture was compared to that of non-CF primary airway ALI cultures (Fig. 1D). Compared to primary cells, iPSC-derived spheroids expressed similar levels of *CFTR*. As expected based on our prior report, *NKX2-1* and *SCGB3A2* were expressed at higher levels in iPSC-derived samples compared to HBEC ALI cultures (Fig. 1D)[33]. Consistent with earlier studies we did not identify mature ciliated or secretory cells in iPSC-derived airway spheroids at this day 28–30 time point. In all, 58 ± 2.7% (mean ± SEM) of cells expressed NKX2-1 at day 28–30 (Fig. 1C). By immunolabeling, CF and non-CF spheroids were composed of NKX2-1+ cells, a subset of which co-expressed TP63 or SCGB3A2 (Fig. 1E). In summary, we generated a bank of CF iPSCs from different classes of CFTR variants and demonstrated the differentiation of CF iPSCs into airway spheroids that express *CFTR* at similar levels to primary airway epithelial cells.

**Characterization of FIS assay in non-CF iPSC-derived airway spheroids.** Through its action on adenylyl cyclase, forskolin activates cAMP-dependent channels, including CFTR[46]. We first sought to determine the CFTR-dependent swelling kinetics in response to forskolin in iPSC-derived cells with normal CFTR (Fig. 2A). Using the same culture conditions as above, non-CF airway spheroids from one iPSC line (non-CF #2; $n = 3$ experiments) were stimulated with 5 μM forskolin or vehicle control (DMSO) and imaged hourly for 24 h (Fig. 2B and Supplementary Movie 1). Individual sphere cross-sectional surface area (CSA) was measured at each time point and calculated as a percentage of initial CSA (Fig. 2B, top panel). Within an experiment, individual spheroids variably swelled (Fig. 2B, top panel; range 112–250%), though average swelling within a well was similar between independent experiments (Fig. 2B, middle panel; range 159–181%). Swelling began within 2 h and continued until hour 20, followed by a plateau phase (20–24 h) (Fig. 2B, middle panel inset). By 24 h, forskolin-stimulated spheroid CSA increased by 173 ± 7% compared to 133 ± 4% (mean ± SEM) for vehicle-treated controls ($P = 0.008$) (Fig. 2B, bottom panel). Further FIS experiments were analyzed after 20–24 h of forskolin exposure.

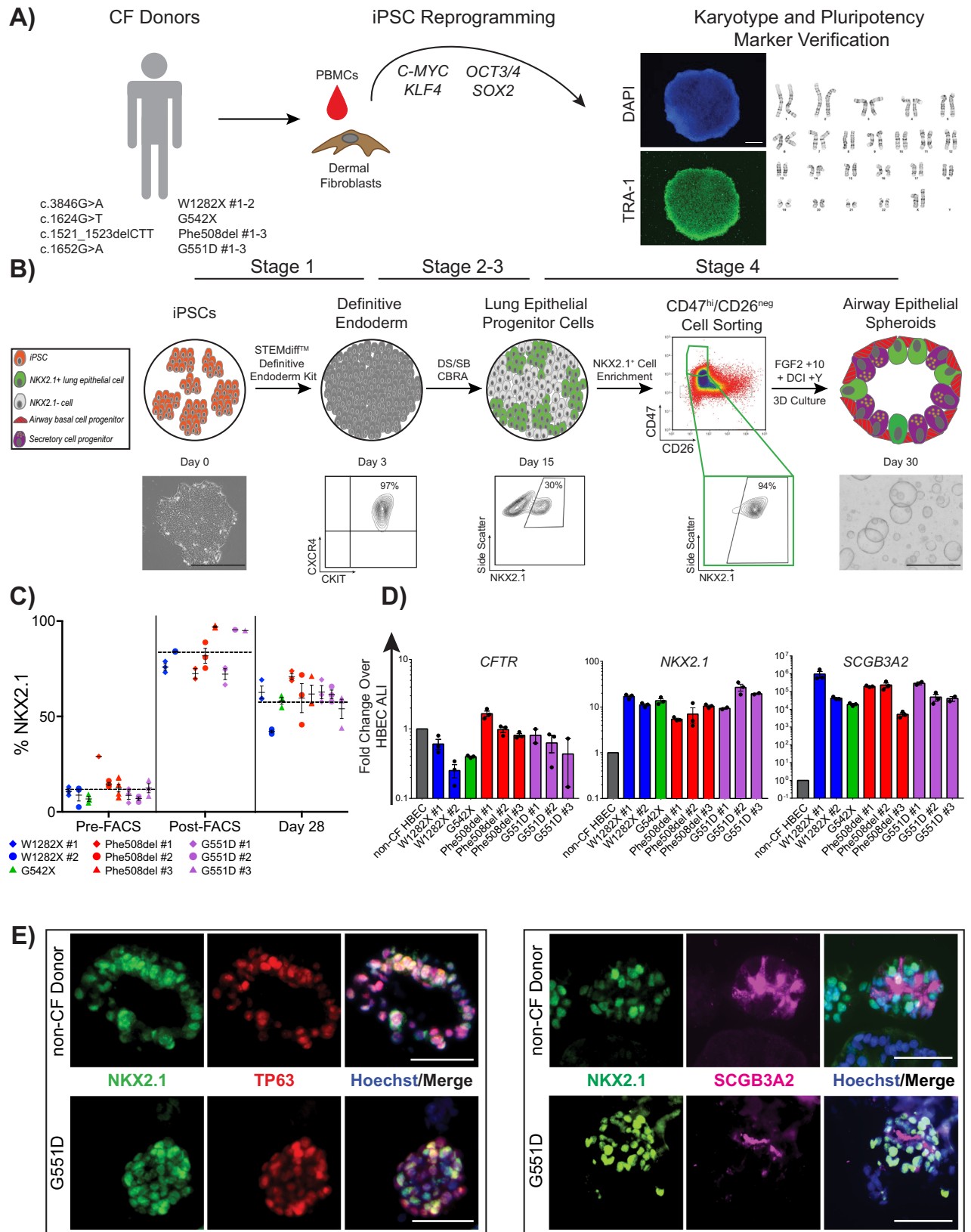

To more broadly assess the forskolin effect between multiple non-CF genetic backgrounds ($n = 3$ cell lines) we generated airway epithelial cell spheroids from the remaining non-CF iPSC lines (non-CF #1, 3) ($n = 3$ experiments per line) (Fig. 2C). Morphologically, non-CF spheroids were heterogeneous in size,

thin-walled, and had a hollow central lumen which increased in size after forskolin treatment (Fig. 2D, insets); average cell numbers per well were similar amongst all samples (Supplementary Fig. 4). We imaged live airway epithelial cell spheroids immediately prior to, and 20–24 h after the addition of forskolin

**Fig. 1 iPSCs derived from multiple CF donors are differentiated into CFTR-expressing airway epithelial cells. A** Schematic describing approach by which selected individuals with CF caused by different classes of variants had somatic cells reprogrammed into iPSCs, which were subsequently tested for markers of pluripotency and G-band karyotype. Variant details and their abbreviations are provided. An example of TRA-1 staining and DAPI nuclear labeling (scale bar represents 500 μm) and normal 46XX karyotype are shown (see Supplementary Fig. 1 for individual karyotyping). **B** Schematic of directed differentiation protocol to generate airway epithelial cell spheroids. Flow cytometry checkpoints shown were utilized to ensure adequate differentiation efficiency at the stages indicated. Representative images of a single iPSC colony and several airway epithelial spheroids shown (scale bars represent 500 μm). **C** Frequency of NKX2-1+ cells as a percentage of all cells at day 14–16, before (Pre-FACS), immediately after sorting CD47$^{hi}$/CD26$^{neg}$ cells (Post-FACS), and 14 days later (day 28). Each point represents an individual experiment for the iPSC line indicated in the key (n = 3 biological replicates of independent differentiations). Horizontal dashed lines represent the average NKX2-1+ % across all cell lines at the designated time point. **D** mRNA expression of canonical airway epithelial cell markers on day 28–30 of differentiation in iPSC-derived airway epithelial spheroids from the iPSC lines indicated, calculated relative to levels within non-CF HBEC-derived ALI cultures. Each point represents an individual experiment (n = 3). **E** Examples of immunolabeling of day 28–30 non-CF and CF iPSC-derived airway epithelial spheroids with antibodies against NKX2-1 and TP63 (left panels) and SCGB3A2 (right panels); scale bars represent 50 μm. Lines and error bars represent mean ± standard error.

or vehicle control (Fig. 2D). Spheroid number and CSA were calculated using automated analytical software (see "Methods") (Fig. 2D)[47]. For each well, we calculated the "Normalized CSA" using the equation: $\frac{Whole\ well\ CSA\ post}{Whole\ well\ CSA\ pre} X100\%$ to assess the mean change in spheroid size after forskolin stimulation. We analyzed 135 ± 26 spheroids per well, 360 ± 77 spheroids for an individual experiment (mean ± SEM) (Fig. 2E), thus on average more than 1000 spheroids for each individual iPSC line. After forskolin stimulation, non-CF spheroid CSA (n = 3 iPSC lines; n = 3 independent differentiations) increased to 155 ± 4%, with no significant difference between genetic backgrounds, compared to 128 ± 4% for vehicle-treated spheroids (P < 0.0001) (Fig. 2F).

Consistent with our prior work identifying some non-lung endoderm emerging on days 28–32, we observed that some cells had lost NKX2-1 expression (Fig. 1C)[45]. We, therefore, assessed the contribution of non-lung (i.e., NKX2-1-) cells to the FIS response. By utilizing the NKX2-1:GFP fluorescent reporter line (non-CF #3), we identified NKX2-1-GFP + and NKX2-1-GFP- spheroids with live-cell fluorescent microscopy before and after forskolin stimulation and found no significant difference between NKX2-1+ and NKX2-1- spheroids (P = 0.60) (Supplementary Fig. 3c). In summary, we developed a platform to consistently measure FIS using non-CF iPSC-derived airway epithelial spheroids from multiple genetic backgrounds.

**Genotype-dependent modulator rescue of CFTR function in a FIS assay.** Having established the baseline FIS response of iPSC-derived airway spheroids from patients with a normal *CFTR* sequence, we next asked if airway spheroids derived from CF donors differed. CF donor iPSCs were differentiated into airway spheroids and characterized as described above. Prior to forskolin stimulation, CF spheroids were smaller and appeared denser than non-CF spheroids (Fig. 3A). Baseline spheroid size varied with the underlying *CFTR* variant such that non-CF spheroids (0.124 ± 0.012 mm$^2$) were larger than G551D (0.090 ± 0.012 mm$^2$), Phe508del (0.047 ± 0.004 mm$^2$), and PTC spheroids (0.016 ± 0.003 mm$^2$) (mean ± SEM) (Fig. 3B).

We next tested baseline FIS of CF spheroids. We analyzed 670 ± 138 spheroids for an individual experiment (mean ± SEM) (Supplementary Fig. 3a). There was small but statistically significant FIS for G551D #1 and #3 spheroids (P = 0.04, P = 0.009) but no detectable swelling in Phe508del spheroids (Fig. 3D), consistent with their expected baseline levels of CFTR function. We next tested the effect of CFTR modulator treatment on CF spheroids (Fig. 3C). VX-770 treatment of G551D spheroids increased the CSA of spheroids to 175 ± 12% of their baseline size (Fig. 3D). Phe508del #1–3 spheroids were treated with currently approved CFTR modulator regimens (VX-809, VX-661, VX-445/VX-661, all in combination with VX-770) (Fig. 3C, D). Treatment

with first-generation correctors (VX-809, VX-661) had a small effect, significant in only 1 of 3 patient lines (Phe508del #1, VX-809 treatment, P < 0.0001; VX-661 treatment, P = 0.0135). However, treatment with the recently approved VX-445 in combination with VX-661/VX-770 led to a robust increase in CSA of all three Phe508del patient cell lines, consistent with known in vitro and clinical effects of VX-445/661/770 (Fig. 3D) (168 ± 11% across three Phe508del lines).

To test the scalability of this platform for potential drug screening, we utilized an automated liquid handler to plate and treat airway epithelial spheroids using the G551D #1 cell line in a 384-well format. Similar to the above experiments, treatment with VX-770 showed a significant CSA increase (155 ± 5.7%) compared to control (111 ± 2.3%) (mean ± SEM) (P < 0.0001) (Supplementary Fig. 5). Thus, we developed a platform that enables the testing of thousands of patient-derived CFTR-expressing airway spheroids across multiple classes of *CFTR* variant with detection of baseline as well as modulator rescue of CFTR function. With the relative ease in generating a large number of airway epithelial cell spheroids and the scalability potential, this platform provides a new tool for patient-specific testing of drug responsiveness and for the development of novel therapeutic strategies.

**Electrophysiologic assessment of CFTR rescue in a polarized iPSC-derived airway epithelium.** The gold-standard platform for the accurate assessment of CFTR function is the measurement of ion transport across differentiated primary human airway epithelium in ALI culture[10]. In our established basal cell differentiation protocol, the aforementioned Phe508del and G551D iPSC-derived airway epithelial cells (Fig. 1B, day 30) were further specified to NGFR+ airway basal cells as previously described (Fig. 4A and Supplementary Fig. 6)[29]. After 8–14 days of ALI culture (Pneumacult-ALI medium), the epithelial layer maintained barrier function (dry apical surface) and transepithelial electrical resistance (TEER) was 887 ± 189 Ω cm$^2$ (mean ± SEM) (n = 3 cell lines) (Supplementary Fig. 6). We observed the presence of motile cilia and mucus hurricanes, a phenomenon similar to primary HBEC ALI cultures (Supplementary Movie 2). When compared to non-CF HBEC ALI cultures, iPSC-derived cells expressed similar levels of relevant airway markers including *NKX2-1*, *SCGB1A1*, *FOXJ1*, *MUC5AC*, and *MUC5B*; as previously described, *SCGB3A2* was expressed at higher levels (Fig. 4B)[29]. Notably, *CFTR* was expressed at similar levels to primary controls. Immunolabeling of ALI cultures confirmed the epithelium was composed of secretory (MUC5AC + ), multiciliated (acetylated-α-tubulin + ) and basal cells (KRT5 + ) (Fig. 4C and Supplementary Fig. 8a, b).

To ascertain the cell type(s) expressing *CFTR*, we re-analyzed a recent single-cell RNA-sequencing (scRNA-seq) dataset from

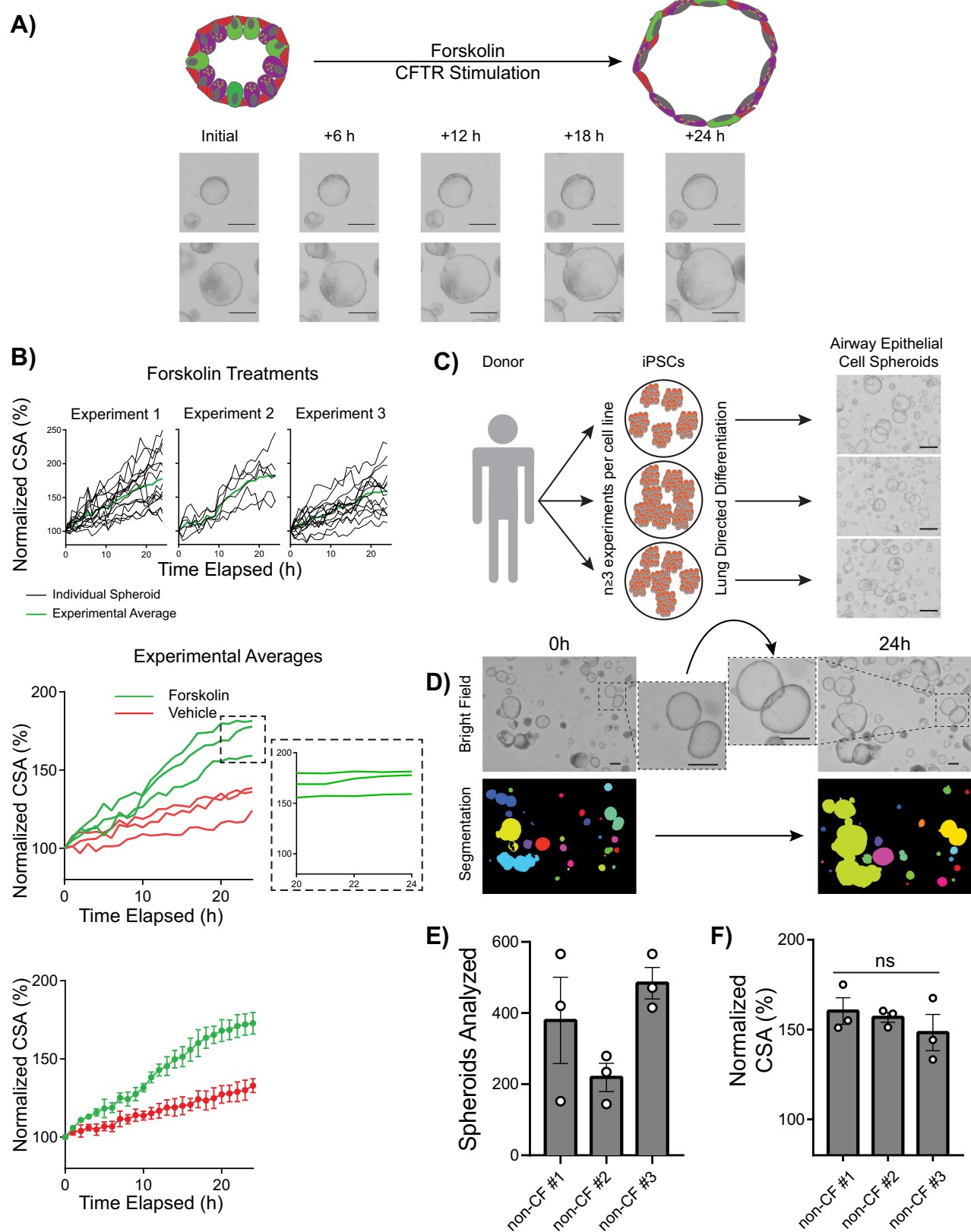

non-CF #3 iPSC-derived mucociliary cells, which were derived using a nearly identical iPSC-directed differentiation protocol[29]. In our prior analysis, Louvain clustering identified six distinct clusters of airway epithelial cells, annotated as follows: (1)

intermediate cells, (2) secretory cells, (3) immature multiciliated cells (MCCs), (4) MCCs, (5) proliferative cells, and (6) basal cells. We found that the majority of *CFTR* transcripts were expressed in cluster 2 (secretory), where the top differentially expressed genes

**Fig. 2 Quantification of FIS in iPSC-derived airway epithelial spheroids from non-CF donors. A** Schematic of FIS assay with representative images of two spheres immediately prior to as well as 6, 12, 18, and 24 h after forskolin addition. **B** Kinetics of non-CF airway spheroid FIS measured by the change in cross-sectional area (CSA). Top panel shows individual spheroid FIS (black lines) and average FIS (green lines) ($n = 3$ experiments). The middle panel shows compiled FIS (and vehicle control) experimental averages with magnified inset indicating 20–24 h. Bottom panel shows mean and standard error for each time point depicted in the middle panel ($n = 3$ experiments). **C** Experimental approach for the FIS assay. Each iPSC line was differentiated in three independent experiments and FIS performed on day 28–32 ($n = 3$ biological replicates of independent differentiations were analyzed per donor for each of three donors). Representative images of non-CF airway epithelial spheroids (day 28) are shown on the right. **D** Automated imaging analysis of non-CF spheroids before (left) and after (right) forskolin stimulation using OrganoSeg™ to quantify the change in CSA. **E** Number of airway spheroids analyzed per experiment for the non-CF cell line indicated. Each point represents an independent experiment. **F** Three non-CF donor FIS responses shown. Each point represents an independent experiment. No significant differences between samples, by one-way ANOVA. Scale bars represent 250 μm. Lines and error bars represent mean ± standard error.

(DEGs) included *SCGB1A1, SCGB3A2, MUC5B*, similar to recent reports from primary airway datasets suggesting the majority of *CFTR* transcripts in human airways are in secretory cells[48,49].

We next tested whether Phe508del iPSC-derived ALI cultures displayed similar electrophysiologic characteristics and modulator rescue to that of published primary HBEC results. In well-differentiated iPSC-derived ALI cultures (Phe508del #1–3) with adequate TEER (see "Methods"), cells were pre-treated with VX-809, VX-661, VX-445/661, or vehicle control. We measured equivalent current ($I_{eq}$) at baseline and after (1) ENaC inhibition, (2) forskolin stimulation, (3) CFTR potentiation, and (4) CFTR inhibition (Fig. 4E, F)[50]. Baseline current ranged from 2.5 to 13 μA/cm$^2$ without significant decrement following ENaC inhibition, similar to previous findings and suggesting low ENaC activity (Fig. 4E, F and Supplementary Fig. 7)[29]. Treatment with CFTR modulators led to significant improvement in forskolin-stimulated current and CFTR inhibition in all Phe508del samples, though magnitude varied between individual lines (Fig. 4F and Supplementary Figs. 6 and 7). Comparison between CFTR modulators showed higher CFTR-specific currents after treatment with the newest generation modulators (12.9 ± 0.5 μA/cm$^2$), with a varied magnitude between cell lines (Fig. 4F). Overall, we determined that iPSC-derived airway cells from Phe508del CF donors can generate mucociliary cultures that express *CFTR* and recapitulate the electrophysiologic baseline and pharmacologic rescue of CFTR-specific current with known modulators. Similarly, we generated ALI cultures for G551D #2–3 and detected significantly increased CFTR current after treatment with VX-770 (Supplementary Fig. 8a–f).

**Assessment of CFTR function in CFTR class 1 iPSC-derived airway epithelial cells**. We next applied both the FIS assay of 3-D spheroids and electrophysiologic assessment of ALI cultures to measure CFTR function in iPSC-derived airway epithelial cells with PTC variants (W1282X #1–2, G542X). Similar to above, we generated airway epithelial cell spheroids and iPSC-derived mucociliary cultures. In the indicated experiments, NKX2-1+ lung epithelial progenitors were purified by sorting for cells expressing the surface marker carboxypeptidase M (CPM) instead of CD47$^{hi}$/CD26$^{neg}$ [44]. Treatment of W1282X #1 airway spheroids using VX-809, VX-661, and/or VX-770 led to no swelling of W1282X spheroids as expected (data not shown). As no targeted treatments are currently available for individuals with class 1 variants, we tested experimental compounds including a read-through agent (G418) and an inhibitor of the NMD pathway (SMG1i) in combination with CFTR modulators. Inhibition of SMG1 was recently demonstrated to increase *CFTR* mRNA and protein quantity, as well as chloride current in W1282X expressing cells[18,19]. In both iPSC-derived airway spheroids and mucociliary cultures, treatment with a SMG1i for 48 h in the basal medium increased *CFTR* mRNA levels by 3.6 ± 0.28 and 3.1 ± 0.07-fold, respectively (mean ± SEM) (Fig. 5A). We next measured FIS of airway spheroids at baseline and in response to treatment with

SMG1i, the aminoglycoside G418, and VX-445/VX-661/VX-770 to measure functional improvement in CFTR activity (Fig. 5B). Treatment with CFTR modulators alone or in combination with G418 did not lead to a significant FIS response; however, combinatorial treatment with VX-445/VX-661/VX-770, G418 and SMG1i led to a significant increase of FIS in W1282X #1 (117 ± 2% of baseline size, $P = 0.0261$) (Fig. 5B). W1282X #2 and G542X were treated identically, however, neither showed improvement in FIS (Fig. 5B and Supplementary Fig. 8g). Finally, we generated mucociliary ALI cultures from W1282X #1 iBCs. Similar to cultures generated from other healthy and CF iPSC lines, we observed an intact airway epithelium composed of basal, secretory, and multi-ciliated cells (Fig. 5C). To compare iPSC-derived cultures with primary cells, we obtained rare HBECs from the explanted lungs of a separate individual, also homozygous for the W1282X variant and generated mucociliary cultures using well-established protocols (see Methods). We tested the electrophysiologic response of both iPSC- and HBEC-derived mucociliary cultures at baseline and after treatment with combinations of CFTR modulators, SMG1i and G418. Minimal ENaC-dependent current was detected in iPSC samples. In both iPSC- and HBEC-derived cultures, there were similar levels of CFTR-dependent current and patterns of responses to treatment combinations (Fig. 5E, F). In both cell types, treatment with the combination of G418, SMG1i, and modulators led to significant improvement in peak delta forskolin current (6 ± 0.85 and 4 ± 0.2 μA/cm$^2$) and CFTR-inhibited current (4.6 ± 0.37 and 4.8 ± 0.42 μA/cm$^2$), with no significant differences between iPSC- and HBEC-derived results (Fig. 5E, F). We treated G542X iPSC-derived ALI cultures (Fig. 5D) with the identical combinatorial therapy, and found a small but statistically significant improvement in CFTR-dependent current (Fig. 5G, H).

## Discussion

Utilizing iPSCs from individuals with class 1–3 *CFTR* variants, we generated *CFTR*-expressing airway epithelial cells and measured baseline as well as the rescue of CFTR channel function with approved and experimental therapeutics. We adapted two established assays of CFTR function—FIS of 3-D spheroids and electrophysiologic assessment of polarized mucociliary epithelia. In both assays, we measured CFTR function and detected modulator rescue in class 2–3 variants. In addition, we have used these platforms to demonstrate the feasibility of identifying and developing potential novel agents for class 1 *CFTR* variants such as W1282X and G542X. To our knowledge, this is the first report of an iPSC-based airway model that can detect genotype-specific CFTR modulator responses. The iPSC-based system has a number of key advantages compared to current in vitro models including the ability to produce large numbers of disease-relevant *CFTR*-expressing airway cells from individuals with rare variants without the need for an invasive biopsy. We anticipate that this platform will be a useful tool in the pursuit of novel therapeutic

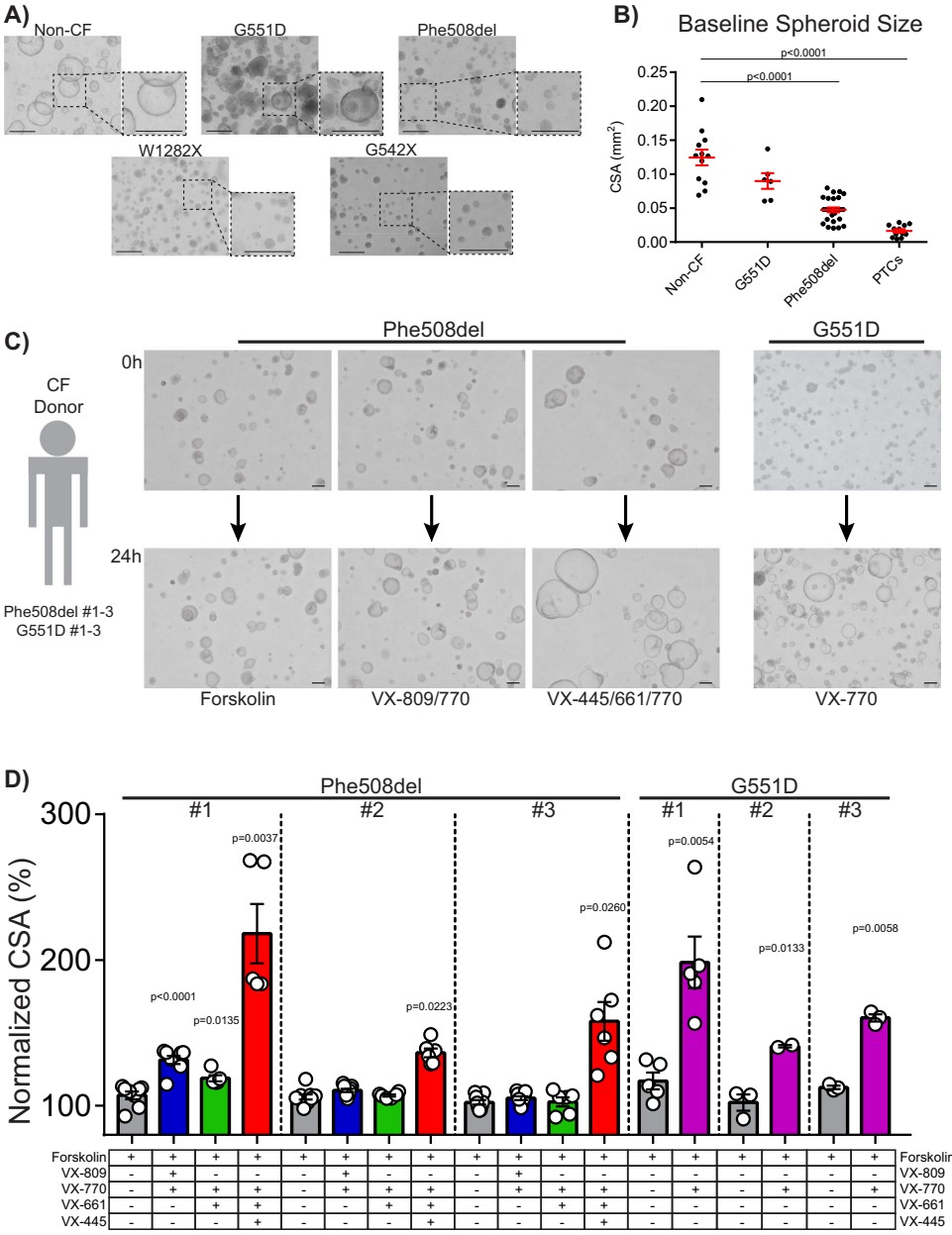

**Fig. 3 Characterization of CF iPSC-derived airway epithelial spheroids in terms of morphology, size, and FIS with and without CFTR modulator treatment. A** Representative microscopy images (of $n \geq 3$ biological replicates from independent differentiations for each cell line) of airway spheroids from CF and non-CF donors demonstrate differences in morphologic appearance; dashed boxes depict magnified views. **B** Baseline spheroid size differs with *CFTR* genotype. Each point represents average spheroid size before forskolin stimulation within an individual experiment (biological replicates from independent differentiations: $n = 12$ for non-CF, $n = 6$ for G551D, $n = 26$ for Phe508del, $n = 12$ for PTCs). **C** Representative spheroid images (of $n > 3$ biological replicates from independent differentiations for each cell line) for the *CFTR* variants indicated before (0 h) and after (24 h) treatment with compounds shown at bottom of panel. **D** Change in CSA after treatment of CF airway epithelial cell spheroids with CFTR modulators ($n > 3$ biological replicates from independent differentiations with exception of $n = 2$ for G551D #2 VX-770 treatment). *P* values were calculated using paired two-tailed Student's *t* test comparing the treatment sample to forskolin control. Scale bars represent 250 μm. Lines and error bars represent mean ± standard error.

approaches for those individuals with CF who continue to struggle without targeted treatment.

The platform developed here builds on the field's recent progress in the directed differentiation of iPSCs into functional *CFTR*-expressing airway epithelial cells[27–29,33,36,42]. First, we expanded on prior proof-of-concept studies using iPSC-derived airway spheroids, composed of immature basal-like and secretory-like cells, which established the CFTR-dependent swelling response to forskolin[33]. We determined the FIS kinetics in iPSC lines with normal CFTR and then measured the FIS in

class 1–3 CF iPSC-derived cells. This overall approach is reminiscent of the initial description of FIS in 1992, when McCray et al. first reported the CFTR-dependent swelling of fetal lung explants in response to cAMP activation using forskolin[46]. More recently, this approach was comprehensively and elegantly applied to primary rectal organoids[20–22]. Some of the major advantages of the rectal organoid system compared to other in vitro platforms include the relative ease of obtaining tissue and its potential scalability. Similar to the published rectal organoid data, iPSC-derived airway epithelial spheroids respond to

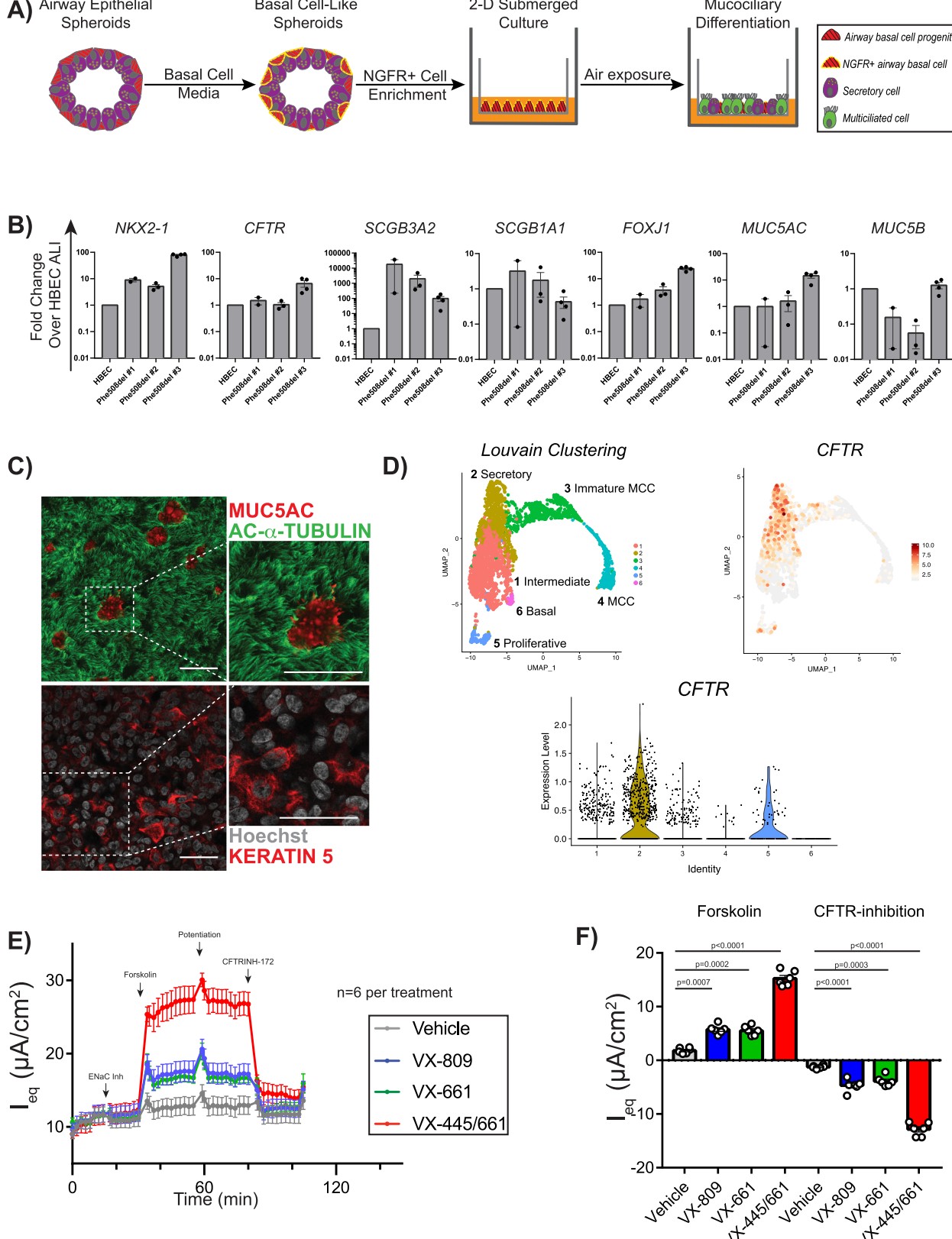

forskolin in a CFTR-dependent manner and also demonstrate pharmacologic rescue of FIS in class 2 and 3 CF spheroids after treatment with CFTR modulators[20]. While we detected significant swelling in non-CF spheroids by two hours, this is notably slower than rectal organoids which swell more rapidly and to a greater size, likely due to higher CFTR expression in rectal epithelial cells[20]. The lower CFTR expression of airway spheroids may also explain the low FIS in response to first-generation correctors (VX-809 or VX-661) in Phe508del samples, however, these molecules have relatively small clinical effects[6,9,51].

**Fig. 4 iPSC-derived airway epithelial cells grow in ALI culture and can be used to detect CFTR-dependent ion flow. A** Schematic of the generation of iPSC-derived ALI cultures via FACS-purified NGFR + airway basal cells. **B** mRNA expression of canonical airway epithelial markers within iPSC-derived ALI cultures, compared to non-CF HBEC ALI cultures (biological replicates from independent differentiations: $n = 2$ for Phe508del #1, $n = 3$ for Phe508del #2, $n = 4$ for Phe508del #3). Each point represents an individual experiment. **C** Example of immunolabeling of iPSC-derived Phe508del ALI cultures with antibodies against markers of multiciliated (acetylated-$\alpha$-tubulin), mucus secreting (MUC5AC), and airway basal cells (KRT5) with nuclei counterstained with Hoechst. **D** Uniform Manifold Approximation and Projections (UMAPs) of scRNA-seq of non-CF iPSC-derived airway ALI cultures depicting Louvain clustering and violin plots demonstrating the expression of *CFTR* in the annotated clusters. **E** Equivalent current measurements of Phe508del ALI cultures ($n = 6$ per treatment). Measurements are shown at baseline and after ENaC inhibition, forskolin treatment, CFTR potentiation with Genistein, and CFTR inhibition (arrows). **F** Quantification of peak delta forskolin and CFTR-inhibition effects of assay results shown in (**E**) ($n = 6$ experimental replicates from independent wells of a differentiation). *P* values were calculated using paired Student's *t* test. Lines and error bars represent mean ± standard error. Scale bars represent 50 μm.

Electrophysiologic measurement of CFTR-dependent ion flux in CF HBEC cultures is a fundamental tool in CF research. However, access to primary HBECs from individuals with rare variants, who are a major research priority, is a significant bottleneck. We sought to determine the fidelity of iPSC-derived mucociliary airway epithelial cultures and found that iPSC-derived cells exhibited CFTR-dependent currents and pharmacologic rescue at magnitudes comparable to primary HBECs[52,53]. For example, Gentzsch et al. measured CFTR-dependent currents of 3–7.5 μA/cm² after treating homozygous Phe508del HBECs with VX-809 ($n = 3$ donors)[53] and Capurro et al. recorded 10–15 μA/cm² CFTR-dependent currents in Phe508del HBECs after VX-445 treatment ($n = 5$ donors)[52]. In iPSC-derived airway cells that generated well-differentiated mucociliary epithelia, VX-809 and VX-445/VX-661 treatment improved CFTR-specific current by 1.3–4.7 and 1.4–12.9 μA/cm², respectively. We noted variability amongst the three different Phe508del iPSC lines and further work will be required to determine if there are donor-specific differences. It should be noted that several quality control steps were included in our differentiation protocol and only iPSC-derived ALI cultures with confirmed robust mucociliary differentiation and acceptable TEER measurements were included for these electrophysiological studies. Despite this and for unclear reasons, iPSC-derived cultures displayed a number of differences including minimal ENaC-dependent current and reduced durability (Supplementary Fig. 7 for raw tracings). Since passage number and culture media can affect the cellular composition of primary airway epithelial cultures, including CFTR-expressing cell types, further work will also be focused on identifying optimal culture conditions for consistent electrophysiological assessments of CFTR function in iPSC-derived cultures[53]. Rare CFTR-rich cells were previously identified in the airways and more recently defined as pulmonary ionocytes[54–56]. While their exact role and contribution to CFTR function is under investigation, studies of freshly isolated human airway epithelium suggest that secretory cells are the major contributor of CFTR mRNA and function[49]. Based on the lack of expression of *FOXI1* mRNA (data not shown), ionocytes are unlikely to be contributing to CFTR function in the iPSC-derived airway epithelial cells studied here.

In a proof-of-concept study to assess the potential of iPSC-derived airway cells to detect small changes in CFTR-dependent current in class 1 variants, we compared mucociliary cultures generated from iPSCs and HBECs from separate donors homozygous for the W1282X variant. We identified (1) small but significant CFTR restoration with combinatorial treatment including G418, SMG1i, and CFTR modulators and (2) an overall similar magnitude of electrophysiologic response between iPSC- and HBEC-derived cultures. We acknowledge the small nature of this study with only two individuals, however, the rarity of primary W1282X HBECs is a major limiting factor in these experiments. This work builds on early proof-of-concept studies using iPSCs to measure CFTR function and more recent work advancing this

technology through fluorescent reporter-based assays to enable high-throughput screens and preclinical drug development pipelines[33,37,42,57,58].

There are several cell-based models of CF, including HBECs, rectal organoids, primary airway organoids, and nasal epithelial cells, each with their own advantages and disadvantages[10,59]. This iPSC platform has several potential advantages. The ability to derive an array of cell types from different organs affected by CF from iPSCs may facilitate studies focused on the multisystem nature of CF. For example, in addition to the airway cells described here, CFTR-expressing intestinal epithelial cells[37], pancreatic exocrine cells[60], and cholangiocytes[61] have been described. Gene editing of iPSCs (as demonstrated with the "C17" genetic background) is a powerful research tool with wide-ranging possibilities. We demonstrated this approach to study effects of specific CFTR mutations while controlling for genetic background (Supplementary Fig. 9b). In the future, this approach may be expanded to determine the role of genetic modifiers in CF and to develop in vivo gene-editing approaches. Since iPSC-derived airway basal cells can be efficiently cryopreserved for long-term storage while retaining their capacity to form CFTR-expressing airway epithelium in established protocols, this platform lends itself to creating biobanks of CF iPSC-derived airway basal cells that can be shared with the worldwide research community[29]. The ambitious approaches aimed at treating or even curing individuals with class 1 variants include complex pharmacotherapy, gene-editing/delivery or cell-based therapies and will require ready access to cellular platforms. While primary HBECs remain the gold-standard assay for CF, iPSC-derived airway cells may accelerate this research by overcoming the bottleneck in accessing human cells carrying rare CFTR variants with ultimate validation of candidate therapeutics in HBECs. Ongoing research in molecular therapeutics includes targeting NMD inhibition (as partially shown here), read-through of PTCs, development of novel CFTR modulators, synthetic transfer RNAs, and modified CFTR mRNAs, all of which may be tested using iPSC-derived airway cells[62]. In this regard, a read-through reagent (ELX-02) was recently demonstrated to have in vitro efficacy in G542X rectal organoids and is currently being evaluated clinically[63]. In vivo gene correction has the potential to cure CF but requires multidisciplinary expertise to accomplish delivery of efficacious but safe gene-editing machinery to the cell type(s) of interest in the airway epithelium. Finally, cell-based therapy for the lungs using either autologous primary or iPSC-derived airway cells is in its infancy, but also has the theoretical potential to cure CF[64]. We anticipate that the iPSC platform described here will be a useful tool, complementary to existing cell-based models of CF, in advancing these endeavors.

We note a number of limitations to the iPSC technology that will require further work. The airway differentiation protocol (like many other iPSC protocols) is lengthy and at each patterning stage, we detect variability. To offset this, we have developed

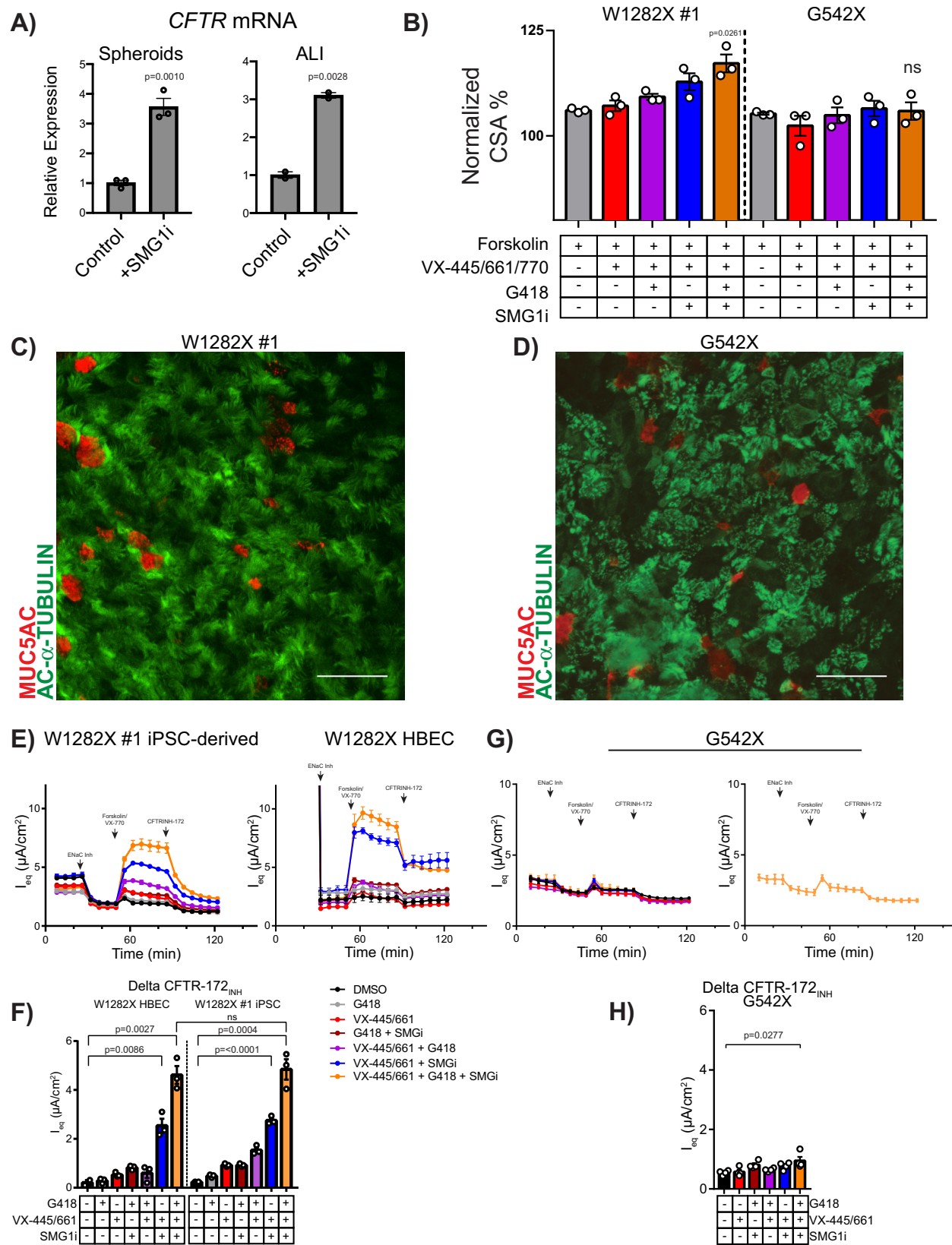

surface marker approaches to isolate first the lung progenitor cells of interest and later iPSC-derived airway basal cells. The reproducibility of this approach is demonstrated through the successful generation of airway epithelial cells in 12 iPSC lines analyzed in this manuscript. While non-lung endodermal cells are detected in airway spheroid cultures, we concluded that the FIS assay was not

affected adversely by their presence. We anticipate that directed differentiation protocols will be further refined in time with a better understanding of the early stages of human lung development. Future studies are needed to determine the extent to which iPSC-derived airway cultures compare to genetically matched primary cells to predict an individual's disease severity and

**Fig. 5 Assessment of CFTR function in W1282X and G542X iPSC-derived airway spheroids and mucociliary cultures. A** Treatment with SMG1i increases mRNA expression levels of *CFTR* in W1282X iPSC-derived spheroids (left) and mucociliary ALI cultures (right) compared to controls (experimental replicates from independent wells of a differentiation: $n = 3$ for control, $n = 2$ for SMG1i treatment). **B** FIS assay of W1282X #1 and G542X iPSC-derived airway spheroids after combination treatment with VX-445/VX-661/VX-770, G418, and SMG1i ($n = 3$). **C**, **D** Examples of immunolabeling of iPSC-derived W1282X #1 and G542X ALI cultures with antibodies against markers of multiciliated (acetylated-α-tubulin) and mucus secreting (MUC5AC) cells. **E**, **F** Electrophysiological assessment and quantification of W1282X #1 iPSC-derived (left) and primary mucociliary cells (right) using a TECC24 instrument after the indicated treatments ($n = 3$ experimental replicates from independent wells of a differentiation). **G**, **H** Electrophysiological assessment and quantification of G542X iPSC-derived mucociliary cells with the indicated treatments. Right panel shows treatment condition that led to improved CFTR current ($n = 3$ experimental replicates from independent wells of a differentiation). *P* values were calculated using unpaired Student's *t* test. Scale bars represent 50 μm. Lines and error bars represent mean ± standard error.

response to CFTR modulators, as the current studies were not powered to do so.

In conclusion, we describe here a platform using iPSC-derived airway epithelial cells that enables the detection of baseline CFTR function and can measure the degree of CFTR rescue by modulator compounds in two distinct assays. The future of CF treatment, particularly for individuals with less common *CFTR* variants, will depend on readily available patient-derived *CFTR*-expressing cells for use in preclinical testing; this iPSC platform offers several potential advantages that will complement existing cell and animal models of CF.

## Methods

**iPSC reprogramming/maintenance**. All experiments involving the differentiation of human iPSC lines were performed with the approval of the Institutional Review Board of Boston University. All iPSC lines used in this project (BU1, BU3, RC2 204, CFTR4-2, C17, CFTR5-5, P20801) were reprogrammed from donor somatic cells (PBMCs or dermal fibroblasts). Reprogramming was performed using a polycistronic lentiviral vector (EF1a-hSTEMCCA4 loxp) (BU1, BU3, RC2 204) or Sendai virus (C17, CFTR4-2, CFTR5-5, P20801, W1282X #2, G551D #3). The iPSC clones—"RC2 204", "C17", "BU1", "BU3"—have been previously described and published[29,39,41,42]. The human embryonic stem-cell line RUES2 was a generous gift from Dr. Ali H. Brivanlou of The Rockefeller University and the iPSC line P20801 was a generous gift from Dr. Scott Diamond of the University of Pennsylvania. All PSC lines were confirmed to have a normal karyotype by G-banding and expressed the expected markers of pluripotency by immunostaining or flow cytometry (Supplementary Fig. 1). Initially grown on inactivated mouse embryonic feeder cells (MEFs), iPSCs were adapted to culture on hESC-Qualified Matrigel (2-D Matrigel) (Corning, catalog #354277) coated tissue culture plates in StemFlex (ThermoFisher, Cat #A3349401) (BU3) or mTESR1 (StemCell Technologies, catalog #85850) (all other cell lines) for maintenance medium. Gentle Cell Dissociation Reagent (StemCell Technologies, catalog #100-0485) was used to passage of PSCs. For further details and description of cell lines, please see https://stemcellbank.bu.edu/Catalog/Item/Home.

**Lung-directed differentiation**. PSC differentiation into NKX2-1+ lung epithelial cell progenitors was carried out as previously reported by our laboratory[29,33,65]. Briefly, PSCs are dissociated to single cells and passaged 24 h prior to differentiation. Cells are then differentiated to definitive endoderm using the STEMdiff Definitive Endoderm Kit (StemCell Technologies, catalog #05110), with 24 h of supplement MR and CJ, followed by 48 h of supplement CJ only. Cells were then treated with Gentle Cell Dissociation Reagent and passaged as small clumps (~20–30 cells) onto a pre-coated (2-D Matrigel) tissue culture plate at dilutions between 1:3 and 1:6. Cells were then treated with "anteriorization" media consisting of complete serum-free differentiation media ("CSFDM") supplemented with 2 μM dorsomorphin (Stemgent, catalog #04-0024) and 10 μM SB-431542 (Tocris, catalog #1614); for the first 24 h, 10 μM Y-27632 (Tocris, catalog #1254) was also included. After 72 h, "specification" media (CBRa) was added for 8-11 days, containing 3 μM CHIR99021 (Tocris, catalog #4423), 10 ng/mL recombinant human BMP4 (R&D Systems, catalog #314-P), and 100 nM retinoic acid (Sigma, catalog #R2625), in CSFDM base. Between days 13-16, a representative well was dissociated and analyzed for "pre-sort" NKX2-1+ cell frequency (intracellular NKX2-1 or GFP flow cytometry, using FlowJo version 10.7.1). The remaining wells were then purified with fluorescence-activated cell sorting (FACS) using cell surface markers CD47hi/CD26neg, CPM[44], or NKX2-1GFP for identification and purification of NKX2-1+ cells, as previously described[28,65].

**Lung epithelial progenitor cell sorting**. Between days 13 and 17, cells were dissociated into single cells using 0.05% trypsin-EDTA (Gibco, catalog #25300062) at 37 °C for 10–12 min. Cells were then washed in an equal volume of "Stop" medium, which contained 10% fetal bovine serum (Gibco, catalog #16141079), and

centrifuged at 300×*g* for 5 min. For cell lines containing NKX2-1:GFP reporter ("C17" and "BU3"), cells were directly resuspended in "Sort buffer," composed of Hanks Balanced Salt Solution, HEPES, EDTA, FBS, and Y-27632, and passed through a 40-μm cell strainer (Fisher, catalog #087711). In all, 10 μM calcein blue AM (Life Technologies, catalog #1429) was added and the cells were sorted for NKX2-1GFP using a high-speed cell sorter (MoFlo Astrios Cell Sorter). Cell lines without NKX2-1:GFP reporter ("BU1", "RUES2", "RC2 204", "CFTR4-2", "P20801", "CFTR5-5," "W1282X #2", "G551D #3") were resuspended in "Sort buffer" at a concentration of 1×10⁶ cells/100 μL and stained with anti-CD47-PerCP Cy5.5 (Biolegend, catalog #323110) and anti-CD26-PE antibodies (Biolegend, catalog #302706) (or anti-CPM antibody) at 4 °C for 30 min. The cells were then washed with 1% fetal bovine serum, centrifuged at 300×*g* for 5 min, and resuspended in "Sort buffer" containing 10 μM calcein blue AM, prior to cell sorting for CD47hi/CD26neg cells as previously described[28].

**Airway epithelial cell specification**. Sorted NKX2-1+ cells were centrifuged at 300×*g* for 5 min, then resuspended in 3-D growth factor reduced Matrigel (Corning, catalog #356231) at a density of 400 cells/μL. In total, 25–50 μL droplets were placed on pre-warmed (37 °C) tissue culture plates, allowed to solidify (10–15 min at 37 °C), then "2 + 10+DCIY" media added to each well, containing FGF2 (R&D Systems, catalog #233-FB), FGF10 (R&D Systems, catalog #345-FG), 50 nM dexamethasone (Sigma, catalog #D4902), 0.1 mM 8-Bromoadenosine 3',5'-cyclic monophosphate sodium salt (Sigma, catalog #B6272), 0.1 mM 3-Isobutyl-1-methylxanthine (Sigma, catalog #I5879), and 10 μM Y-27632, in base CSFDM. Media was changed every 48–72 h for the next 14–16 days.

**Assessment of airway epithelial cell spheroids (days 28–30)**. A representative droplet was dissociated using 2 mg/mL dispase (Gibco, catalog #17105041) at 37 °C for 30–45 min, to allow the dissolution of the 3-D Matrigel. Multicellular spheroids were washed with 1% fetal bovine serum and centrifuged at 200 *g* for 3 min. The pelleted spheroids were resuspended in 0.05% trypsin-EDTA at 37 °C for 10–12 min until a single-cell suspension was observed. The cells were washed with 10% fetal bovine serum and centrifuged at 300×*g* for 5 min. NKX2-1:GFP reporter-containing cell lines were then assess with flow cytometry for NKX2-1GFP cell frequency using either BD FACSCaliber or Stratedigm S1000EON. PSC lines not containing the NKX2-1:GFP construct were fixed with 1.6% paraformaldehyde (Alfa Aesar, catalog #43368) for 10 min at 37 °C, then permeabilized with Intracellular Staining and Permeabilization Wash Buffer (Biolegend, catalog #421002) and stained with rabbit anti-TTF1 antibody (Abcam, catalog #EP1584Y; 1:500) for 30 min at room temperature. Cells were centrifuged at 300×*g* for 5 min, then stained with donkey anti-rabbit-Alexa488 antibody for 30 min protected from light. Cells were centrifuged at 300×*g* for 5 min and assessed by flow cytometry for % NKX2-1+ cells. NKX2-1- cells (either NKX2-1GFP- or CD47low) were used for negative controls and flow cytometry gating. For quality control, only experiments with >60% NKX2-1+ cells at this stage were used for downstream analysis.

**FIS assay**. After confirmation of adequate NKX2-1 retention on days 28–30, 2–4 droplets were dissociated as above with 2 mg/mL dispase. Spheroids were then washed 2–3 times with 1% fetal bovine serum and centrifuged at 200×*g* for 3 min. Spheroids were then resuspended in fresh 3-D Matrigel (4 °C) at a low density to prevent significant spheroid overlap, so as to not impede downstream imaging and analysis. Resuspended spheroids were plated as droplets on pre-warmed (37 °C) tissue culture plates, coated with 2 + 10+DCIY containing media, and allowed to recover for 48–72 h. 2 + 10+DCIY media was supplemented with the indicated treatment or vehicle control. CFTR modulators (VX-809, VX-661, VX-770, VX-445) (Selleckchem catalog #S1565, #S7059, #S1144, #S8851), were reconstituted with DMSO and used at a final concentration of 3 μM; G418 (Selleckchem, catalog #S3028) was reconstituted with tissue culture grade water at a final working concentration of 10 μM; SMG1i was a generous gift from the Cystic Fibrosis Foundation, and was reconstituted with DMSO at a final working concentration of 0.5 μM. Spheroids were pre-treated (Phe508del, W1282X, G542X lines) with CFTR correctors, G418, SMG1i or vehicle for 24 h. After 24 h, high-resolution spheroid images (z-stack 15-μm thickness; brightfield) were captured using a Keyence BZ-X700 fluorescence microscope. Spheroids were then immediately fed with fresh

2 + 10+DCIY media supplemented with indicated treatment and 5 μM forskolin (Sigma, catalog #F3917) or vehicle control. After 20–24 h, repeat images were captured and analyzed identically to the pre-forskolin images above. Saved images (.tif) were analyzed using OrganoSeg[TM], an open-source MatLab plug-in that has been previously described[47]. To capture the cross-sectional area of all organoids per well (hereafter "whole-well CSA"), images were segmented with the following settings: intensity threshold 0.5; window size 100; size threshold 100. For quality control, each post-segmentation image was reviewed manually to exclude (1) spheroids on edge, (2) spheroids that had burst or flattened, and (3) spheroids not included on both pre- and post-images. OrganoSeg[TM] output for each image included total sphere number and CSA. "Normalized" whole-well CSA was compared between pre- and post-forskolin images using the equation:
$\frac{Whole\ well\ CSA\ post}{Whole\ well\ CSA\ pre}X100\%$. Each experiment included the treatment conditions shown in figures as well as a control well. Control wells were treated with DMSO (vehicle for forskolin, modulators, and SMG1i) and/or tissue culture grade water (vehicle for G418). The CSA measured in control wells was subtracted from all other CSA measurements.

Within each independent experiment, at least three replicate wells were performed in parallel per treatment. We analyzed an average of 592 spheroids per treatment group (per experiment) amongst all cell lines (Supplementary Fig. 3).

For time-lapse studies, spheroids were treated with forskolin or vehicle control (similar to above) and placed in an environmental chamber within the Keyence BZ-X700 fluorescence microscope. For the entirety of the assay (24 h), chamber temperature and $CO_2$ content were monitored and maintained at 37 °C and 5%, respectively. Images were automatically captured each hour, then analyzed for change in CSA with time. Manual spheroid CSA measurement was performed using ImageJ (version 1.0).

**Airway basal cell generation.** NGFR + airway basal-like cells were differentiated as previously described[29]. Briefly, days 28–30 airway epithelial cells were dissociated from 3-D Matrigel droplets and into single-cell suspension as described above. Cells were resuspended in fresh 3-D Matrigel at 400 cells/μL and re-fed with 2 + 10+DCIY medium for 24 h, after which the medium was changed to "Basal Cell Medium," composed of PneumaCult-Ex Plus (StemCell Technologies, catalog #05040) supplemented with small-molecular inhibitors of TGF-beta (1 μM A83-01; Tocris, catalog #2939) and BMP signaling pathways (1 μM DMH1; Tocris, catalog #4126) as well as 10 μM Y-27632. After 10–14 days, 3-D Matrigel was dissolved and a single-cell suspension was obtained as described above. Cells were resuspended in "Sort buffer" at $1 \times 10^6$ cells/100 μL and labeled with mouse monoclonal APC-conjugated anti-NGFR antibody (Biolegend, catalog #345108) for 30 min at 4 °C protected from light. Mouse monoclonal IgG1k APC-conjugated antibody (Biolegend, catalog #400122) was used in a subset of cells for isotype control and gating. 10 μM calcein blue AM was added to samples prior to sorting for live NGFR:APC-positive cells. Sorted cells were centrifuged at 300×g for 5 min and resuspended in "Basal Cell Medium" at $6 \times 10^4$ cells/100 μL. In total, 100 μL cell suspension was added to the apical chamber of 6.5-mm Transwell inserts (Corning, catalog #CLS3470) or 24-well TECC24 plates (Corning, catalog #CLS3396) pre-coated with 2-D ESC-qualified Matrigel or recombinant human laminin-521, as well as 600 μL "Basal Cell Media" to the basolateral chamber. Cells reached 70–90% confluence (3–7 days) before the medium was transitioned to commercially available PneumaCult-ALI (StemCell Technologies, catalog #05001); 24 h after medium change, the apical chamber was aspirated. Every 48–72 h, the basolateral chamber medium was changed and the apical chamber was washed with 100 μL PBS (10 min at 37 °C). To assess the integrity of the epithelial layer, transepithelial electrical resistance (TEER) was measured 7–10 days after apical chamber aspiration using a Millicell ERS-2 Volt-ohm meter (Millipore catalog #MERS00002). Each well (including a 2-D Matrigel-coated blank well containing no cells) was fed with fresh PneumaCult-ALI medium (600 μL basolateral; 200 μL apical) and resistance (in Ohms) measured between apical and basolateral chambers. All measurements were done in triplicate and then averaged for each well ($R_{total}$). The resistance of the blank well ($R_{blank}$) was subtracted out, then net resistance ($R_{tissue}$) adjusted for surface area of the insert (0.3 cm²). The equations used are as follows: 1) $R_{tissue} = R_{total} - R_{blank}$ and 2) $TEER(\Omega x cm^2) = R_{tissue}(\Omega)x\ surface\ area(cm^2)$. ALI cultures with TEER values less than 100 Ω cm² were discarded.

**ALI culture electrophysiology.** Functional electrophysiology was measured using protocols adapted from published literature[19,50]. Briefly, the transepithelial electrical resistance and the equivalent current ($I_{eq}$) were recorded using a 24-channel transepithelial current clamp amplifier (TECC24, EP Design, Bertem, Belgium). To determine the baseline resistance, initial measurements were collected at 2–5 minute intervals. Changes in $I_{eq}$ were determined after the addition of Benzamil, Forskolin (FSK), CFTR potentiator (Genistein (Gen) or VX-770 as indicated), and a mixture of CFTR inhibitor-172, GlyH-101, and Bumetanide, as described. TECC24 assays were performed on days 14–21 after apical chamber aspiration.

**Primary human bronchial epithelial cell culture.** For HBEC RNA samples, early passage (P2) primary HBEC cultures were grown in ALI culture conditions as

previously described[50]. They were expanded and allowed to differentiate while exposed to air apically until day 14. Cells were dissociated into single cells and collected in QIAzol lysis reagent (Qiagen, catalog #79306) and stored at −80 °C until RNA purification. W1282X/W1282X HBEC cultures used for electrophysiologic analysis (Fig. 5D–F) were similarly early passage (P2) and grown in standard published conditions[50].

**RT-qPCR.** Cell samples ($5 \times 10^4$–$1 \times 10^5$ cells) were collected at the indicated day in QIAzol lysis reagent and stored at −80 °C or used immediately for RNA purification. RNA was isolated using the Qiagen RNeasy mini kit (catalog #74014). cDNA was synthesized via reverse transcription of 150 ng purified RNA, as previously described (Applied Biosystems, catalog #98080234)[28]. For qPCR, we performed technical triplicates for each cDNA product (1:4 diluted). In total, 10 μL or 20 μL reactions were performed using Applied Biosystems QuantStudio7 384-well or Applied Biosystems StepOne 96-well System, respectively, for 40 cycles. Primers used were Taqman probes as indicated in Supplementary Table 1. Relative gene expression was calculated based on the average cycle ($C_t$) of the technical triplicates, normalized to 18 S control, and presented as fold change ($2^{-\Delta\Delta Ct}$) and compared to the primary control indicated. Undetected Taqman probes were assigned a $C_t$ value of 40.

**Immunostaining.** Day 28–30 airway epithelial spheroids were dissociated from 3-D Matrigel as above, fixed using 4% paraformaldehyde (PFA) for 4 h at room temperature, then embedded in paraffin after dehydration with ethanol and clearing with xylene, as previously described[33]. In all, 5-μm sections were obtained from paraffin blocks using an HM 325 Rotary Microtome (ThermoFisher Scientific), then removed from paraffin with washes of xylene, and increasingly dilute ethanol. Antigen retrieval was performed using a citric acid-based antigen unmasking solution (Vector laboratories, product H-3300).

ALI cultures were washed with PBS and fixed with 4% PFA for at least 15 min. Samples were permeabilized with 0.1% Triton-X solution (Sigma Aldrich, T9284) and blocked with 4% normal donkey serum (NDS) (Sigma Aldrich, D9663). Samples were stained with primary antibodies (Supplementary Table 1) overnight at 4 °C, washed the following day, then stained with conjugated secondary antibodies and Hoechst 33342 nuclear counterstain (ThermoFisher H3570), as indicated. Prolong Gold antifade (ThermoFisher) was added, the coverslip was placed, and the samples were dried overnight prior to imaging. Microscopy of fixed, immunolabeled cells was performed using either a Leica DMi8 Confocal microscope (Leica Microsystems, Wetzlar, Germany) and Leica Application Suite Software (Leica Microsystems) or a Nikon Eclipse Ni-E microscope (Melville, NY).

**Single-cell RNA-sequencing.** Single-cell RNA-sequencing was previously performed using non-CF #3 ("BU3")[29]. Briefly, single-cell reads were aligned to the reference genome GRCh38 to obtain a gene-to-cell count matrix with Cell Ranger version 3.0.2 (10x Genomics). Pre-processing was performed with Seurat (version 3.1.0), then non-linear dimensionality reduction was performed using Uniform Manifold Approximation and Projection (UMAP) for visualization purposes. Clusters identified were annotated based on their top differentially expressed genes. *CFTR* expression was compared between known, previously annotated clusters as indicated.

**Gene editing of "C17" iPSC line.** Oligonucleotides were purchased from Integrated DNA Technologies (IDT). SNP discrimination assays were custom ordered from IDT or purchased from ThermoFisher Scientific [I507del/WT assay: C__151693868; G551D/WT assay: C__32545127]. Guide RNA (gRNA) sequences, ssODN sequences, sequencing primers, genotype controls, and SNP discrimination assays for G542 (WT) >G542X and W1282 (WT) >W1282X editing were the same as previously described, as was the SNP discrimination assay for F508del/WT[19]. gRNA, ssODN donor, and sequencing primers for G551D correction are in Supplementary Table 2. Correction of C17-NKX2-1[GFP] (a gift from Brian Davies) from CFTR Ile507del/Phe508del to WT/WT was achieved in one editing step using the two guides and 1 of 3 ssODNs shown in Supplementary Table 2. pSpCas9(BB) −2A-GFP (PX458) was a gift from Feng Zhang (Addgene plasmid # 48138; http://n2t.net/addgene:48138; RRID:Addgene_48138) and gRNA sequences were cloned according to the previously published protocol[66].

Cells were nucleofected using the Lonza 4D-Nucleofector system using the Lonza P3 Primary Cell 4D-Nucleofector X Kit L and program CB-150. Each reaction contained ~1 million cells, 25–50 ſg total of pX458 plasmid(s) with appropriate guide(s) and HDR donor plasmid (P63-mcherry insertion) (if applicable), and 2 ſL of 100 ſM ssODN HDR donor, if applicable. pX458-GFP + iPSC cells were isolated by FACS (GFP +) and ~100 subclones were generated. Clonal lines were screened for the desired edit using the SNP discrimination assays in qPCR format followed by Sanger sequencing as previously described[19]. Correction of "C17" (Ile507del/Phe508del >WT/WT) was screened independently with Phe508del/WT and Ile507del/WT SNP discrimination assays and preliminary hits had WT, but not mutant signal in both assays.

To insert the TP63[mCherry] reporter, a donor plasmid consisting of a Furin/GSG/P2A, and mCherry coding sequence with three PAM-inactivating mutations, with

homology arms was used as the HDR donor to insert this cassette just prior to the stop codon of P63 (Supplementary Table 3).

**Statistical analysis**. Unless otherwise indicated, two-tailed Student's *t* tests were applied for statistical comparison, as mentioned in figure legends. When appropriate, paired comparisons were performed between groups; "*n*" used for statistical testing represents the number of independent experiments. A *P* value <0.05 was considered to indicate a significant difference between groups.

**Reporting summary**. Further information on research design is available in the Nature Research Reporting Summary linked to this article.

## Data availability

The data presented in this manuscript are available from the authors upon reasonable request. The accession number for the scRNA-seq dataset shared in this manuscript is GSE142246. Source data are provided with this paper.

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

## Acknowledgements

We are indebted to Anne Hinds for technical support with embedding and sectioning of spheroids; Brian Tilton for expertise in cell sorting; Sam Gallant for technical support in electrophysiology; Carlos Villacorta-Martin for his expertise in single-cell RNA-sequencing analysis. We are indebted to Greg Miller and Marianne Janes for overall laboratory support as well as for reprogramming and characterization of iPSC lines. We sincerely thank Darrell Kotton for his guidance, support, and vision of this project and research center. iPSC lines (as indicated) were generously supplied by the University of Pennsylvania, Cystic Fibrosis Foundation, and The Hospital for Sick Children (University of Toronto). This project was supported by T32:HL7035-44 (A.B.), CFF BER-ICA2010 (A.B.), T32:HL7035-44 (J.A.L.), NIH grant DK065988, (S.H.R.), CFF BOUCHE19RO (S.H.R.), CFF RANDEL20XX2 (S.H.R.), Emily's Entourage (S.H.R. and F.J.H.), R01 HL139799-04 (F.J.H.), and CFF HAWKIN20XX2 (F.J.H.). All authors critically evaluated and approved of the manuscript.

## Author contributions

A.B. and F.J.H. conceived the work, designed the experiments, analyzed the data, and wrote the manuscript. A.B., J.A.L.S., M.L.B., J.L., A.M., A.S., and M.P. performed differentiation experiments. R.L., J.L., J.H., K.C., J.M., and S.R. performed the electrophysiology experiments and provided valuable discussion. K.B., R.S., and H.V. performed gene-editing and fluorescent reporter targeting. N.R. and D.T. performed the differentiation and experiments utilizing the automated liquid handler. G.M., K.H., and P.M. provided valuable discussion and patient samples.

## Competing interests
