## [Peer Review File · Nature Communications]

A multimodal iPSC platform for cystic fibrosis drug testingREVIEWER COMMENTS

Reviewer #1 (Remarks to the Author):

In their manuscript “A multimodal iPSC platform for cystic 1 fibrosis drug testing” Bercal et al. describe the utilization of disease specific iPSCs from patients suffering from CF. The major findings include:

- 1) A panel of induced pluripotent stem cells (iPSCs) derived from individuals with common or rare CFTR variants were applied to measure CFTR function in iPSC-derived airway cells.
- 2) CFTR function was measured in iPSC-derived spheroids and planar ALI cultures using two established assays (forskolin-induced swelling and electrophysiological measurement of CFTR dependent current).
- 3) Responses to known CFTR modulators were measured and genotypic differences (patient specific) were detected.

In principle, the experiments have been well designed and the presented data appear sound.

To my opinion, however, the study does not contribute much additional novelty to justify publication in a prestigious journal like Nat Communications.

There are various papers starting with two reports in Nat Med and Cell Stem Cell already in 2013 that utilize patient specific intestinal organoids derived from intestinal biopsies. Based on the initial reports of the Clevers group the effects of available drug and drug combinations on various mutations in patient-derived intestinal organoids have been investigated. When exploring and comparing different CFTR mutations, the intestinal organoids technology seems to have clear advantages compared to iPSC technology, as derivation of intestinal organoids is less laborious, time consuming and expensive than generation and differentiation of patient specific iPSCs.

Also, long term expanding airway spheroids derived from airway brushes have been used in CF research (Sachs, Embo J 2019).

Protocols for targeted airway differentiation and enrichment of lung progenitors from hiPSCs that are described in the manuscript are not novel and have already been published by the same group.

Also the use of iPSC technology in CF research has already been reported:

There are various reports describing the generation of iPSC lines from CF patients that carry different classes of CFTR mutations.

Such cells have been differentiated into several CF-relevant cell lineages (airway, intestinal, pancreas). Gene correction of these cells (e.g. Fleischer et al., 2020) as well as targeted introduction of different transgene reporters that facilitate functional read-out have been described. Even a high throughput screen based on patient-specific iPSCs has already been conducted (Merkert et al., 2019).

In summary, I don't think that the presented findings represent a sufficiently striking advance to justify publication in Nature Communications

Reviewer #2 (Remarks to the Author):

This is an exciting research paper entitled "A multimodal iPSC platform for cystic fibrosis drug testing" by Berical et al., which builds on many previous publications using human cystic fibrosis (CF) iPSC-derived lung models for therapeutic drug screens (Jiang et al., 2021; Ahmadi et al., 2017; Wong et al., 2012 and work by the same group McCauley et al., 2017). Of course, as improvements in differentiation protocols emerge, the ability to generate robust, renewable sources of patient-derived cells will provide a reliable model for testing existing and emerging new therapeutics.

CF is a multiorgan disease and recent developments in new "CFTR modulators" has led to promising therapies for some, but not all, CF patients. Importantly, patients with rare CF mutations such as Class I production mutants (ie. W1282X) are not eligible for current CFTR modulators and therefore the efficacy of these new drugs remains unknown for many patients harbouring rare mutations. Having a patient-derived cell source to test the effectiveness of new compounds as a preclinical screen will be beneficial and support clinical decisions in prescribing these expensive but potentially life-saving drugs to patients.

Here Berical et al., demonstrates the utility of iPSC-derived 3D spheroids and polarized airway epithelial cells in air liquid interface (ALI) harbouring either homozygous mutations for Class I (1 W1282X), Class II (3 F508del, one line being a compound het variant, F508del/Ile507del) and Class III (1 G551D) in modeling the response to CFTR modulators in-vitro. The authors tested several known CFTR modulators (correctors and potentiator) and compounds (SMGi and G418) to determine genotype-specific responses and demonstrate CFTR function as a measure of forskolin-induced swelling (FIS) in spheroids or transepithelial electrical resistance (TEER). Importantly, primary human bronchial epithelial cells (not from the same patient source) were used to benchmark iPSC-derived responses. Overall, this is a great demonstration of the functional utility of iPSC derived models and the potential of these models for precision medicine.

Strengths

1. This work leverages a new differentiation protocol established by the same group (Hawkins et al., 2021) and enriches for lung NKX2-1+ progenitor cells using cell surface isolation for CD47hiCD26neg (Hawkins et al., 2017). This new protocol efficiently generates basal stem cells capable of multi-lineage epithelial differentiation and importantly as shown here, can generate 3D spheroids and polarized epithelial in ALI. They now show the use of these iPSC-derived lung models as a new tool for testing existing compounds aimed at targeting the CF genetic mutation which is timely (given the new developments in new generation modulators) and showcase the power of iPSC-derived models for

future precision medicine. Importantly, while current modulators are approved from F508del (>90% of the CF population), historically, drug responses have varied from patient to patient (likely due to gene modifiers, environment etc). Therefore having a robust preclinical cell model to predict patient responses is an exciting and promising avenue for iPSC-based tools.

2. The use of technical replicates and in this case at least 3 sets of differentiations that show similar outcomes is important. One of the concerns with differentiations is the reproducibility of the differentiations and this was certainly an issue in the early days when lot-to-lot differences in growth factors had a large impact in the efficiency of differentiations. Nonetheless, to show meaningful results from different sets of differentiations is important. The difference between lines (in this case from different sources) is expected and may reflect the heterogeneity observed in-vivo in the patient population.

-Limitations

While I am enthusiastic about this work, there are some concerns that needs to be addressed.

1. While I can accept the n=3 F508del lines used, the n=1 for the other two mutations is not sufficient to make a general conclusion about genotype-specific responses. Have the authors tried differentiation additional iPSC lines from the same “patient” at the very least to determine reproducibility of the results? There is a well-known rich biobank at the Toronto Hospital for Sick Children (<https://lab.research.sickkids.ca/cfit/>) that has now generated >100 CF iPSC lines from patients (Eckford et al., 2018) with a range of CF-disease causing mutations. There are at least 5 W1282X CF iPSC lines in this bioresource and 2 of which have gene-corrected isogenic controls to allow benchmarking functional responses at the “individual” level (as each person’s “normal” might vary, due to other factors). Moreover, this biobank also has primary patient nasal cells, a good surrogate for HBEC for patient-specific benchmarking. I do encourage use of additional lines for the other two classes of mutations.

2. The similarity in FIS in NKX2-1+ versus NKX2-1 neg spheroids is concerning. If all the lines contained the NKX2-1GFP reporter, this may not be an issue as one would just measure changes in NKX2-1 expressing spheroids. But in the CF iPSC-derived lines, the only method for “enriching” NKX2-1 cells (~70% NKX2-1+) cells is using the CD47/CD26 strategy. This means there are likely at least 30% NKX2-1neg spheroids in their FIS assay. How does the authors account for a NKX2-1neg spheroid affecting the results? And more importantly, what are these NKX2-1 neg spheroids? Is the swelling in NKX2-1 neg spheroids because of CFTR? or alternative channels that can also be stimulated through forskolin and thus bias the results?

3. The spheroids look cystic in structure and does not have the thick bilayer epithelium that is typically observed in spheroids derived from primary bronchial cells (ie. Rock et al., 2009). It’s a bit strange because the organoids (Fig 1E) appear to be almost exclusively made of TP63 basal stem cells. While SCGB3A2 is expressed in the lumen of the spheroid, can the authors use a different marker to show secretory cells make up the spheroids? Especially since in Fig 4D, it is understood that secretory cells

express the majority of CFTR in their system (albeit that was ALI cultures), not basal cells. Finally, is the reason for the cystic phenotype of the spheroid because of the limited cell type in the spheroid? Assuming these are “immature” spheroids and not equivalent to “mature” HBEC cells they used, then the similarity in response between iPSC-derived and HBEC-derived spheroids is rather surprising.

4. An issue that appears to be a hurdle for all cell-based models is the ability to predict patient-specific responses. Previously, Dekkers et al., used primary rectal organoid swelling in response to CFTR modulators to demonstrate the ability of these organoids to predict patient in-vivo responses. In this work, are these iPSC-derived airway models also a good indicator of “in-vivo” lung function responses? Along the same line, can the authors comment on whether their model captures “patient-specific” response? The use of primary cells such as HBEC to benchmark responses is ok but is it really a true standard knowing that there are clearly heterogeneities in disease phenotype and responses to therapy amongst patients with the same CF genetic mutation?

5. There does not seem to be a correlation between spheroid FIS and TEER of ALI cultures. Specifically, FIS in CFTR modulator treated spheroids in F508del showed significant swelling in all three F508del lines with new modulators 445/661 but the TEER of ALI cultures from the same lines only showed effect for 1 of 3 lines. Can the authors explain the discrepancy? And how does this impact development of the screens for personalized therapy? Which model is right?

Minor concerns

1. Ionocytes were previously identified to be a major source of CFTR expression (Plasschaert et al., 2018; Montoro et al 2018). There does not appear to be ionocytes in these cultures? Is this due to an inefficiency in generating ionocytes or perhaps these ionocytes do not emerge from basal stem cells as had previously been suggested by Montoro et al?

2. Can the authors comment on why ciliated cells do not appear to express CFTR (Fig 4D)?

3. Were the correctors in the media when forskolin and the potentiators (Gen or VX-770) added?

Reviewer #3 (Remarks to the Author):

Dear Authors,

The authors present a well written (apart from the CFTR abbreviation in the abstract) and documented study on the use of iPSC for CF modeling. In general, it shows what you would expect to find so that is good. The impact is somewhat reduced as iPSC based airway models for CF and CFTR modulator testing has been shown previously. Still, in my opinion, the current paper addresses a relevant need which is the development of protocols that enable larger scale testing in airway(-like) cells.

Major point:

- The study would strengthen significantly by adding more patient genotypes to better show the performance of the assay across more CFTR genotypes and samples.
- Fig 3D: are these three replicates from a single one day experiment or multiple experiments? The main point is how strong can you show that at different culture timepoints the donor-to-donor differences are maintained?

Minor points:

- To my opinion, it is a bit too easy to say that rectal organoids may not recapitulate (airway) disease because of the intestinal origin and lack of pathological features (the gut is also strongly affected in CF, and rectal biopsies have long been used for diagnostic testing). It is like saying that the sweat glands are also not important while it represents a main diagnostic classifier and outcome parameter for drugs. It has been shown that intestinal organoids capture FEV1 and SCC responses to treatment. Also relations with other clinical features have been shown and a current MedrXiv study of the Beekman group would also strongly substantiate this (but not peer reviewed yet). Other studies with rectal tissue (e.g. Marcus Mall) also show relations with disease severity. Albeit that HBE are a golden standard in the field for preclinical research, I don't think that any in vitro cell model in CF has been as strongly validated as biomarker for individual disease features or therapeutic response as intestinal organoids have been. I think this could be discussed in a bit more balanced manner. The data points out that in general, CFTR biology across epithelial tissues appears sufficiently preserved to be important.

The authors also highlight that rectal organoids may not be used/represented for class I mutations which I not find particularly strong. Work in organoids correctly predicted in the past that PTC124, a read-through agent for class I (without efficacy in organoids), would not be clinically efficacious as found later in phase 3 studies. At that time, G418 responses in primary airway epithelium were not yet published to my knowledge. So the statement that organoids may not represent class I therapies appears somewhat out of context. Recent published work also showing ELX2 efficacy in rectal organoids, and others showed NMD in rectal organoids (group of Margarida Amaral).

But obviously, the authors have a point that different cellular backgrounds may affect responses and for this reason their work is very relevant. But in general the overlap between CFTR function measurements

in HBE and intestinal organoids is actually pretty good. For other epithelial function, the airway systems is clearly better (e.g Sachs et al shows that airway organoids and intestinal organoids differ in swelling in response to Ca^{2+} -inducing agonists).

There is definitely a lot of potential use for iPSC based protocols as presented here. I think the use of this for basic studies and drug development can be very important and likely. I find a case to use iPSC as a living diagnostic for CF difficult, as current intestinal or nasal cell models provide a quicker and already validated tool. On a whole, I would recommend to bring a bit more detail into the proposed uses and comparisons between different models.

We thank the reviewers for their careful consideration of our manuscript. We now include a revised manuscript. The main feedback from Reviewers 2-3 was that increasing the number of genotypes would strengthen the conclusions and the manuscript. While identifying and recruiting new patients to generate and differentiate iPSCs would not have been within a revision timeline we are delighted to report that through the CF iPSC bank at Sick Kids Toronto and through the Cystic Fibrosis Foundation Therapeutics Lab, the revised manuscript now includes 3 Class 1, 3 Class 2, and 3 Class 3 iPSC lines and total of 12 samples. We think this significantly strengthens the conclusions of the manuscript. We hope the Reviewers find this revised manuscript responsive to their concerns. Changes are highlighted in yellow throughout the revised manuscript.

Reviewer #1 (Remarks to the Author):

In their manuscript "A multimodal iPSC platform for cystic 1 fibrosis drug testing" Berical et al. describe the utilization of disease specific iPSCs from patients suffering from CF.

The major findings include:

- 1) A panel of induced pluripotent stem cells (iPSCs) derived from individuals with common or rare CFTR variants were applied to measure CFTR function in iPSC-derived airway cells.
- 2) CFTR function was measured in iPSC-derived spheroids and planar ALI cultures using two established assays (forskolin-induced swelling and electrophysiological measurement of CFTR dependent current).
- 3) Responses to known CFTR modulators were measured and genotypic differences (patient specific) were detected.

In principle, the experiments have been well designed and the presented data appear sound.

To my opinion, however, the study does not contribute much additional novelty to justify publication in a prestigious journal like Nat Communications.

There are various papers starting with two reports in Nat Med and Cell Stem Cell already in 2013 that utilize patient specific intestinal organoids derived from intestinal biopsies. Based on the initial reports of the Clevers group the effects of available drug and drug combinations on various mutations in patient-derived intestinal organoids have been investigated. When exploring and comparing different CFTR mutations, the intestinal organoids technology seems to have clear advantages compared to iPS technology, as derivation of intestinal organoids is less laborious, time consuming and expensive than generation and differentiation of patient specific iPSCs.

Also, long term expanding airway spheroids derived from airway brushes have been used in CF research (Sachs, Embo J 2019).

We thank the reviewer for their comments. While we appreciate that this is not the first report of using the iPSC system in CF we respectfully suggest that this work does represent an advance on the current literature. We have edited the discussion to frame this more clearly and would like to make the following points. 1) There is an ongoing need for effective or curative therapies for patients with CF and this forms the basis of the Cystic Fibrosis Foundation's current Path to a Cure initiative which includes developing cell models and tools to discover, develop, and evaluate the efficacy of

CFTR therapeutics. 2) The patients with non-sense mutations represent a significant bottleneck as their cells are very valuable, rarely obtained, and yet now represent a major research focus. This is the motivation for our work. 3) We agree that the Clevers and Beekman groups have pioneered the important rectal organoid platform and have edited the discussion to better reflect this. 4) Despite their work there is a clear need, documented by experts in the field, for human airway models from patients with rare CFTR mutations to develop future therapies (J.P. Clancy et al., Journal of Cystic Fibrosis, 2018). 5) Sachs et al., described important progress in deriving primary airway organoids, including from 5 patients with CF, all of whom had at least one residual function allele. These organoids are generated from either lung tissue at the time of resection or (in the case of the CF studies) through bronchoalveolar lavage. CF patients have highly inflamed and infected lungs, often with drug resistant organisms, which has long affected the procurement of these cells from lung tissue. As such, we argue that Sachs et al. is certainly an important advance but does not solve the bottleneck in access to tissue from patients with rare mutations nor did they demonstrate class to class differences or include cells from homozygous class I mutations. We hope the reviewer finds the revised discussion frames this more clearly and the addition of new iPSC lines highlights the genotype-specific response.

Protocols for targeted airway differentiation and enrichment of lung progenitors from hiPSCs that are described in the manuscript are not novel and have already been published by the same group.

Also the use of iPSC technology in CF research has already been reported:

There are various reports describing the generation of iPSC lines from CF patients that carry different classes of CFTR mutations.

Such cells have been differentiated into several CF-relevant cell lineages (airway, intestinal, pancreas). Gene correction of these cells (e.g. Fleischer et al., 2020) as well as targeted introduction of different transgene reporters that facilitate functional read-out have been described. Even a high throughput screen based on patient-specific iPSCs has already been conducted (Merkert et al., 2019).

In summary, I don't think that the presented findings represent a sufficiently striking advance to justify publication in Nature Communications

J. P. Clancy et al. (Journal of Cystic Fibrosis, 2018) highlighted the many attractive features of iPSCs for the CF field but concluded that the sensitivity, specificity, positive and negative predictive value of the platform(s) were unknown. The main theme of this manuscript is the logical and necessary progression from published work from our group and others establishing the potential of iPSC technology to detect CFTR function to address these uncertainties. We agree that many groups, including Drs. Snoeck, Spence, Gotoh, Firth, Rajagopal have published the derivation of airway cells from iPSCs. These works are referenced and not claimed to be novel in this manuscript. We and others have reported the generation of iPSCs from CF and we previously reported both the gene-correction of CF iPSCs and proof-of-concept FIS (Crane et al., 2015 and McCauley et al., 2017). Fleischer et al., describe a similar approach except in iPSC-

derived intestinal organoids. Importantly, they performed gene-correction, differentiation, and FIS of cells from a single delF508 donor. Merkert et al., represents an exciting advance in applying previously established fluorescence read outs of CFTR rescue to an iPSC platform enabling high throughput approach. This work used cells from a single delF508 donor. We regret the oversight of not citing this work and have corrected this. Our manuscript addresses different and complementary questions. In response to the Reviewers feedback we have increased the number of donors and genotypes. This represents an important advance in terms of measuring modulator efficacy across genotypes/classes, ensuring feasibility across multiple donors, and including rare non sense CFTR mutations that represent the future research challenge. iPSC-based papers are often, and appropriately, criticized for small sample numbers (given the cost and time required to reprogram and differentiate them). The 12 unique iPSC lines now included in this revised manuscript represents the most robust cohort of iPSC lines used in lung disease modeling to date (to our knowledge). We hope the reviewer finds these additional data convincing and we thank them for their time and expertise.

Reviewer 2 Comments

This is an exciting research paper entitled “A multimodal iPSC platform for cystic fibrosis drug testing” by Berical et al., which builds on many previous publications using human cystic fibrosis (CF) iPSC-derived lung models for therapeutic drug screens (Jiang et al., 2021; Ahmadi et al., 2017; Wong et al., 2012 and work by the same group McCauley et al., 2017). Of course, as improvements in differentiation protocols emerge, the ability to generate robust, renewable sources of patient-derived cells will provide a reliable model for testing existing and emerging new therapeutics.

CF is a multiorgan disease and recent developments in new “CFTR modulators” has led to promising therapies for some, but not all, CF patients. Importantly, patients with rare CF mutations such as Class I production mutants (ie. W1282X) are not eligible for current CFTR modulators and therefore the efficacy of these new drugs remains unknown for many patients harbouring rare mutations. Having a patient-derived cell source to test the effectiveness of new compounds as a preclinical screen will be beneficial and support clinical decisions in prescribing these expensive but potentially life-saving drugs to patients.

Here Berical et al., demonstrates the utility of iPSC-derived 3D spheroids and polarized airway epithelial cells in air liquid interface (ALI) harbouring either homozygous mutations for Class I (1 W1282X), Class II (3 F508del, one line being a compound het variant, F508del/Ile507del) and Class III (1 G551D) in modeling the response to CFTR modulators in-vitro. The authors tested several known CFTR modulators (correctors and potentiator) and compounds (SMGi and G418) to determine genotype-specific responses and demonstrate CFTR function as a measure of forskolin-induced swelling (FIS) in spheroids or transepithelial electrical resistance (TEER). Importantly, primary human bronchial epithelial cells (not from the same patient source) were used to benchmark iPSC-derived responses. Overall, this is a great demonstration of the functional utility of iPSC derived models and the potential of these models for precision medicine.

Strengths

1. This work leverages a new differentiation protocol established by the same group (Hawkins et al., 2021) and enriches for lung NKX2-1+ progenitor cells using cell surface isolation for CD47^{hi}CD26^{neg} (Hawkins et al., 2017). This new protocol efficiently generates basal stem cells capable of multi-lineage epithelial differentiation and importantly as shown here, can generate 3D spheroids and polarized epithelial in ALI. They now show the use of these iPSC-derived lung models as a new tool for testing existing compounds aimed at targeting the CF genetic mutation which is timely (given the new developments in new generation modulators) and showcase the power of iPSC-derived models for future precision medicine. Importantly, while current modulators are approved from F508del (>90% of the CF population), historically, drug responses have varied from patient to patient (likely due to gene modifiers, environment etc). Therefore having a robust preclinical cell model to predict patient responses is an exciting and promising avenue for iPSC-based tools.

2. The use of technical replicates and in this case at least 3 sets of differentiations that show similar outcomes is important. One of the concerns with differentiations is the reproducibility of the differentiations and this was certainly an issue in the early days when lot-to-lot differences in growth factors had a large impact in the efficiency of differentiations. Nonetheless, to show meaningful results from different sets of differentiations is important. The difference between lines (in this case from different sources) is expected and may reflect the heterogeneity observed in-vivo in the patient population.

-Limitations

1. While I can accept the n=3 F508del lines used, the n=1 for the other two mutations is not sufficient to make a general conclusion about genotype-specific responses. Have the authors tried differentiation additional iPSC lines from the same “patient” at the very least to determine reproducibility of the results? There is a well-known rich biobank at the Toronto Hospital for Sick Children (<https://lab.research.sickkids.ca/cfit/>) that has now generated >100 CF iPSC lines from patients (Eckford et al., 2018) with a range of CF-disease causing mutations. There are at least 5 W1282X CF iPSC lines in this bioresource and 2 of which have gene-corrected isogenic controls to allow benchmarking functional responses at the “individual” level (as each person’s “normal” might vary, due to other factors). Moreover, this biobank also has primary patient nasal cells, a good surrogate for HBEC for patient-specific benchmarking. I do encourage use of additional lines for the other two classes of mutations.

Thank you for the helpful suggestion. To address this and to improve the generalizability of the generated data (especially for Class 1 and Class 3 CFTR mutations), we are delighted to report that we were successful in increasing the number and genotypes of donor iPSC lines from a total of 8 to 12. We obtained four additional iPSC lines with homozygous CFTR mutations. Through the generosity of the Cystic Fibrosis Foundation as well as The Hospital for Sick Children (University of Toronto), we have added two iPSC lines with Class 1 mutations (in text: W1282X #2, G542X) as well as two lines for Class III CFTR mutations (in text: G551D #2, #3) (iPSC characteristics shown in

Supplemental Figure 1). We now include data demonstrating the differentiation and CFTR functional analysis of all four of these iPSC lines to airway epithelial cells, in triplicate experiments. Overall, we find consistent responses across genotypes. Briefly, figure 1A has been updated to include the new genetic backgrounds, as well as the differentiation characterizations depicted in 1C-D. Figures 3B (spheroid size) and 3D (G551D #2-3 FIS) have also been updated to reflect the additional data generated from these iPSC lines. As expected for a highly drug-responsive mutation such as G551D, there is a significant improvement in CFTR function (Figure 3D) after treatment with VX-770. The new premature stop codon (PTC) iPSC lines (W1282X #2 and G542X) were also included in the FIS studies and these data are shown in Figure 5B (G542X) and Supplemental Figure 8G (W1282X #2). As expected, neither of these mutations demonstrated a statistically significant increase in FIS (after treatments tested).

We are happy to report and include in the revised manuscript, that iPSC-derived ALI cultures were generated from three of these new donors as well (G551D #2-3, G542X) (Figure 5D; Supplemental Figure 8A,B) and CFTR function was assessed at baseline and in response to modulators by electrophysiologic analysis. As expected, there was a significant response in G551D #2-3 with VX-770 treatment (Supplemental Figure 8C-F) whereas G542X demonstrated a small though statistically significant increase in CFTR function after combination treatment shown (Figure 5G-H). Overall, these additional data, support the claim that genotype-specific responses are observed.

Overall, the Reviewer's comments were very helpful as was the direction to established banks of CF iPSC lines.

2. The similarity in FIS in NKX2-1+ versus NKX2-1 neg spheroids is concerning. If all the lines contained the NKX2-1GFP reporter, this may not be an issue as one would just measure changes in NKX2-1 expressing spheroids. But in the CF iPSC-derived lines, the only method for "enriching" NKX2-1 cells (~70% NKX2-1+) cells is using the CD47/CD26 strategy. This means there are likely at least 30% NKX2-1neg spheroids in their FIS assay. How does the authors account for a NKX2-1neg spheroid affecting the results? And more importantly, what are these NKX2-1 neg spheroids? Is the swelling in NKX2-1 neg spheroids because of CFTR? or alternative channels that can also be stimulated through forskolin and thus bias the results?

The reviewer asks several important questions. We have previously published that early iPSC-derived lung progenitors retain some plasticity and in the early stages of the protocol can revert to a non-lung endoderm phenotype (McCauley et al., 2018 and Hurley et al., 2020). Single-cell transcriptional profiling of these (lung and) non-lung cells on protocol day 27 (and day 41), suggest that they are most similar to hepatocyte-like cells (McCauley et al, 2018; Figures 5-6). Using either an NKX2-1^{GFP} line, CD47^{hi}/CD26^{neg} or CPM sort strategy (as described by Shimpei Gotoh's group), significantly enriches but does not eliminate these cells.. We have previously performed experiments that address the reviewer's questions. In terms of the specificity of the FIS assay for CFTR (rather than alternative ion channels), we previously demonstrated that FIS at this timepoint in all spheroids is CFTR-dependent. As described in McCauley et

al (2017), we gene-corrected one allele of a Phe508del homozygous iPSC line, then compared the corrected and uncorrected lines using the FIS assay (McCauley et al, 2017; Figure 6). In these experiments (done without CFTR modulators), there was no significant FIS in the unedited line; however, gene editing (of only the CFTR allele) demonstrated a rescue of FIS, thus showing the specificity of this assay as a readout for CFTR function.

In this manuscript, due to similar concerns raised by the reviewer, we have analyzed the FIS of both NKX2-1^{GFP+} and NKX2-1^{GFP-} spheroids (Supplemental Figure 3C). While the NKX2-1^{GFP-} spheroids (predominantly hepatic, as described above) express CFTR and swell in response to forskolin, they do so at an almost identical level to NKX2-1^{GFP+} spheroids. As such, this issue does not significantly affect the interpretation of the assay.

3. The spheroids look cystic in structure and does not have the thick bilayer epithelium that is typically observed in spheroids derived from primary bronchial cells (ie. Rock et al., 2009). It's a bit strange because the organoids (Fig 1E) appear to be almost exclusively made of TP63 basal stem cells. While SCGB3A2 is expressed in the lumen of the spheroid, can the authors use a different marker to show secretory cells make up the spheroids? Especially since in Fig 4D, it is understood that secretory cells express the majority of CFTR in their system (albeit that was ALI cultures), not basal cells. Finally, is the reason for the cystic phenotype of the spheroid because of the limited cell type in the spheroid? Assuming these are "immature" spheroids and not equivalent to "mature" HBEC cells they used, then the similarity in response between iPSC-derived and HBEC-derived spheroids is rather surprising.

Yes, the iPSC-derived airway spheroids tend to be "cystic" in appearance and are commonly composed of a single monolayer of cells. The SCGB3A2 immunolabeling shown in Figure 1E (right panel) is to demonstrate the derivation of these cells from CF iPSC lines and shows both cytoplasmic and extracellular (luminal) staining. We are happy to report that we have extensively characterized these cells (McCauley et al., Cell Stem Cell, 2017; McCauley et al., Stem Cell Reports, 2018; Hawkins et al., Cell Stem Cell, 2021). This analysis included gene expression profiling, immunostaining (2017), single-cell RNA-Seq, and use of SCGB3A2 and TP63 reporter lines to profile these cells. Combined, these data demonstrated that spheroids at this time point are composed of TP63+ airway basal cell progenitors and SCGB3A2+ secretory progenitors with some variability but in approximately equal frequencies. These cells have a more immature gene expression profile than mature, post-natal basal and secretory cells. The SCGB3A2+ cells were previously demonstrated to express SCGB1A1 and in fact this was the most highly expressed gene in sorted SCGB3A2+ cells (excluding SCGB3A2) (McCauley 2018). We appreciate the importance of this question given the expression of CFTR in the secretory population. Finally, we speculate that the cystic nature does reflect the more immature nature of the cells based on the observation in Hawkins et al., 2021 that iBCs differentiated in 3-D in a tracheosphere assay (later stage of the protocol) were more reminiscent of primary airway organoids.

4. An issue that appears to be a hurdle for all cell-based models is the ability to predict patient-specific responses. Previously, Dekkers et al., used primary rectal organoid swelling in response to CFTR modulators to demonstrate the ability of these organoids to predict patient in-vivo responses. In this work, are these iPSC-derived airway models also a good indicator of “in-vivo” lung function responses? Along the same line, can the authors comment on whether their model captures “patient-specific” response? The use of primary cells such as HBEC to benchmark responses is ok but is it really a true standard knowing that there are clearly heterogeneities in disease phenotype and responses to therapy amongst patients with the same CF genetic mutation?

This is an excellent point and answering this will be crucial for future applications of these technologies. We know that patients with at least one Phe508del allele have highly variable clinical responses to Trikafta. However, with the relatively small number of individuals and genotypes in the present study, these experiments are not powered to answer this important question. To adequately address this, larger and appropriately powered experiments must be carried out, ideally with comparisons to genetically matched primary airway epithelial cells that are beyond the scope of this current manuscript.

5. There does not seem to be a correlation between spheroid FIS and TEER of ALI cultures. Specifically, FIS in CFTR modulator treated spheroids in F508del showed significant swelling in all three F508del lines with new modulators 445/661 but the TEER of ALI cultures from the same lines only showed effect for 1 of 3 lines. Can the authors explain the discrepancy? And how does this impact development of the screens for personalized therapy? Which model is right?

This is a critical question. The reviewer’s observation centers on the discrepancy between FIS and electrophysiologic results, and which assay best approximates a patient’s clinical response. Firstly, direct comparison of these two assays is challenging for several reasons, such as intrinsic differences in the cells from two distinct timepoints as well as variable metrics of accuracy for each assay. While the specific amplitude of response (between assays) cannot be compared, the reviewer correctly notes that the pattern of response (i.e., VX445/661 > VX-809) is not consistent between platforms. The expected response (VX-445/661 > VX-661 \cong VX-809) is observed for all Phe508del samples in the FIS assay (Figure 3D) but only one of three samples for electrophysiologic testing (Supplemental Figure 6C-D). The reasons for this observation remain unknown, but ongoing work will address possibilities including: 1) level and maturity of CFTR protein in ALI cultures, 2) comparison of results to primary and genetically matched HBECs, and 3) optimal ALI media conditions for upregulation of CFTR expressing cells. The reviewer clearly makes a valid point - to advance this platform and to adequately assess for its effectiveness as a personalized medicine tool, in addition to the above, future work must stress: 1) larger numbers of donors, 2) comparisons of platform responses to in-vivo clinical responses, and 3) the contribution of genetic modifiers with whole exome sequencing (for example) to explain potential donor to donor differences (as published in several studies). Future experiments that

focus on these parameters will help shed light on which platform is right. In terms of which model is right, we suggest that the FIS assay is more suitable for medium throughput screens and the ALI model is more sensitive for small but potentially important CFTR responses.

Minor concerns

1. Ionocytes were previously identified to be a major source of CFTR expression (Plasschaert et al., 2018; Montoro et al 2018). There does not appear to be ionocytes in these cultures? Is this due to an inefficiency in generating ionocytes or perhaps these ionocytes do not emerge from basal stem cells as had previously been suggested by Montoro et al?

This is an excellent observation that required clarification in the manuscript. Ionocytes are rare, highly CFTR-expressing cells recently identified in the mouse and human lung. Recent studies employing high resolution single cell RNA-sequencing of the human lung do indeed identify this small cell population, though the summative majority of CFTR transcripts is shown to exist within secretory cells (Okuda et al, 2021). To the reviewer's point, we routinely screened for FOXI1 mRNA in these experiments and did not detect ionocytes in our iPSC-derived airway epithelial cell cultures. We intermittently detect FOXI1+/CFTR+ cells in ALI cultures generated from higher passage iBCs (not in 3-D spheroids, data not shown) and we are currently investigating these cells. Given this observation, it is likely that ionocytes emerge from a basal cell progenitor population as Montoro et al. described, though more detailed future studies are needed (and underway) to clearly answer this question.

2. Can the authors comment on why ciliated cells do not appear to express CFTR (Fig 4D)?

Until recently, the airway multiciliated cell was thought to contain the majority of CFTR and be a necessary target of CF therapies. Over the last several years, the research field has identified other and more important CFTR expressing cells (such as the ionocyte and secretory cell, as above). Okuda et al 2021 (mentioned above) assessed the single cell RNA-sequencing of human lung samples and demonstrated "low and infrequent" expression of CFTR in ciliated cells, which is similar to our presented data set in Figure 4D. Additionally, recent work shows that the previously reported CFTR expression in ciliated cells was due to antibody cross reactivity (Sato et al., Scientific Reports 11, 23256 (2021)).

3. Were the correctors in the media when forskolin and the potentiators (Gen or VX-770) added?

In both assays (FIS and electrophysiology), the CFTR correctors (VX-445, 661, 809) remain in the media from the time of pre-treatment until the end of the assay. At the time of forskolin and potentiator addition, fresh media is added which contains all treatment compounds, including correctors.

Reviewer 3 Comments

Major point:

- The study would strengthen significantly by adding more patient genotypes to better show the performance of the assay across more CFTR genotypes and samples.

We completely agree and have addressed this suggestion and concern as above (Reviewer 2, Major Comment 1). We think this significantly improves the manuscript.

- Fig 3D: are these three replicates from a single one day experiment or multiple experiments? The main point is how strong can you show that at different culture timepoints the donor-to-donor differences are maintained?

The majority of the data presented with the initial submission represented three separate experiments for an individual genetic background, however in one case there were only two replicates (Phe508del #1: VX-661 and VX-445/661). Roughly half of these experiments were completely separate differentiations (months apart from each other) with the rest reflecting cells that were separated as iPSCs and differentiated synchronously. To address the reviewer's valid concern about inter-experimental variability, we have repeated and now include differentiations and FIS assays for each of the three Phe508del donors, each again in triplicate experiments. The compiled results for all FIS assays are now shown in Figure 3D, and the data are broken down by experiment number (1-3), which were each carried out on a separate date (Supp. Figure 9). Nearly all comparisons were similar between dates, however differences in amplitudes (but not pattern of response) are noted for Phe508del #1 and #3 after VX-445/661/770 treatment (red bars). While important to establish run to run variability, we feel that the utility of this assay is unchanged, as the trend between experiments is consistent (i.e., VX-445/661/770 is always superior to vehicle control, VX-661/770, and VX-809/770).

Minor points:

- To my opinion, it is a bit too easy to say that rectal organoids may not recapitulate (airway) disease because of the intestinal origin and lack of pathological features (the gut is also strongly affected in CF, and rectal biopsies have long been used for diagnostic testing). It is like saying that the sweat glands are also not important while it represents a main diagnostic classifier and outcome parameter for drugs. It has been shown that intestinal organoids capture FEV1 and SCC responses to treatment. Also relations with other clinical features have been shown and a current MedRxiv study of the Beekman group would also strongly substantiate this (but not peer reviewed yet). Other studies with rectal tissue (e.g. Marcus Mall) also show relations with disease severity. Albeit that HBE are a golden standard in the field for preclinical research, I don't think that any in vitro cell model in CF has been as strongly validated as biomarker for individual disease features or therapeutic response as intestinal organoids have been. I think this could be discussed in a bit more balanced manner. The data points out that in general, CFTR biology across epithelial tissues appears sufficiently preserved to be important.

The authors also highlight that rectal organoids may not be used/represented for class I mutations which I find particularly strong. Work in organoids correctly predicted in the past that PTC124, a read-through agent for class I (without efficacy in organoids), would not be clinically efficacious as found later in phase 3 studies. At that time, G418 responses in primary airway epithelium were not yet published to my knowledge. So the statement that organoids may not represent class I therapies appears somewhat out of context. Recent published work also showing ELX2 efficacy in rectal organoids, and others showed NMD in rectal organoids (group of Margarida Amaral).

But obviously, the authors have a point that different cellular backgrounds may affect responses and for this reason their work is very relevant. But in general the overlap between CFTR function measurements in HBE and intestinal organoids is actually pretty good. For other epithelial function, the airway system is clearly better (e.g Sachs et al shows that airway organoids and intestinal organoids differ in swelling in response to Ca^{2+} -inducing agonists).

There is definitely a lot of potential use for iPSC based protocols as presented here. I think the use of this for basic studies and drug development can be very important and likely. I find a case to use iPSC as a living diagnostic for CF difficult, as current intestinal or nasal cell models provide a quicker and already validated tool. On a whole, I would recommend to bring a bit more detail into the proposed uses and comparisons between different models.

We agree with the reviewer's points. The rectal organoids are an important and advanced system and it was certainly not our intention to cast them in a negative or unbalanced manner. We have edited the discussion and hope the reviewer finds the changes more balanced (removed sentence from end of paragraph 3 in the introduction and edited the end of paragraph 2 and beginning of paragraph 3 in the discussion). Changes are shown in yellow highlight.

REVIEWERS' COMMENTS

Reviewer #2 (Remarks to the Author):

As previously noted in my review, this work is an exciting incremental advancement in demonstrating the utility of iPSC-based models for disease modelling and therapy discovery TOWARDS precision medicine. This resubmission is a significantly improved version which the authors have carefully factored in the reviewers concerns. These include adding more patient genotypes/lines to strengthen their observations and conclusions, clarifying discussion points, and acknowledging important relevant work in the field.

I am satisfied with this version of the manuscript and have no new major concerns. Well done and thank you for your careful considerations of the reviewer's concerns.

Reviewer #3 (Remarks to the Author):

My comments have been addressed, paper is suited for publication